# Benchmarking Multimodal LLMs on Recognition and Understanding over Chemical Tables

## Abstract

With the widespread application of multimodal large language models in scientific intelligence, there is an urgent need for more challenging evaluation benchmarks to assess their ability to understand complex scientific data. Scientific tables, as core carriers of knowledge representation, combine text, symbols, and graphics, forming a typical multimodal reasoning scenario. However, existing benchmarks are mostly focused on general domains, failing to reflect the unique structural complexity and domain-specific semantics inherent in scientific research. Chemical tables are particularly representative: they intertwine structured variables such as reagents, conditions, and yields with visual symbols like molecular structures and chemical formulas, posing significant challenges to models in cross-modal alignment and semantic parsing. To address this, we propose ChemTable—a large-scale benchmark of chemical tables constructed from real-world literature, containing expert-annotated cell layouts, logical structures, and domain-specific labels. It supports two core tasks: (1) table recognition (structure and content extraction); and (2) table understanding (descriptive and reasoning-based question answering). Evaluation on ChemTable shows that while mainstream multimodal models perform reasonably well in layout parsing, they still face significant limitations when handling critical elements such as molecular structures and symbolic conventions. Closed-source models lead overall but still fall short of human-level performance. This work provides a realistic testing platform for evaluating scientific multimodal understanding, revealing the current bottlenecks in domain-specific reasoning and advancing the development of intelligent systems for scientific research.[1]

## 1 Introduction

Recent advances in multimodal large language models (MLLMs) have created new opportunities for mining expert knowledge from scientific literature and are increasingly viewed as catalysts for AI-driven scientific discovery (Zhang et al., 2024). A growing wave of *scientific accelerators*—such as MLLMs-based OCR, ChatPaper (Dean et al., 2023), and ChatPDF (Panda, 2023)—demonstrates the potential of MLLMs to literature parsing, summarization, and interactive reading. From a modeling perspective, MLLMs exhibit strong capabilities in semantic understanding and reasoning, making them well-suited for processing the rich multimodal content of scientific documents. Benchmarks like ChartQA (Masry et al., 2022) and ChartX (Xia et al., 2024) have begun exploring visual reasoning over scientific figures. Yet, scientific literature remains one of the most semantically dense and domain-specialized corpora, and understanding it serves as both a valuable application and a rigorous testbed for evaluating the limits of MLLMs (Li et al., 2024b). In particular, while MLLMs excel at general-purpose visual tasks, they continue to struggle with domain-specific multimodal reasoning, precisely the capability that underpins **AI-assisted scientific discovery**.

While recent benchmarks have primarily focused on figures and charts, **tables remain a largely underexplored yet equally critical modality in scientific literature**. In chemistry, tables are concise and information-rich representations of experimental setups, reaction conditions, and empirical results. These tables often combine symbolic expressions, structured variables, and graphical elements—posing significant challenges for existing MLLMs. Despite their importance, there is a lack of realistic, domain-specific datasets designed to evaluate the capabilities of MLLMs in scientific table

---

[1] https://anonymous.4open.science/r/ChemTable-ICLR-2026

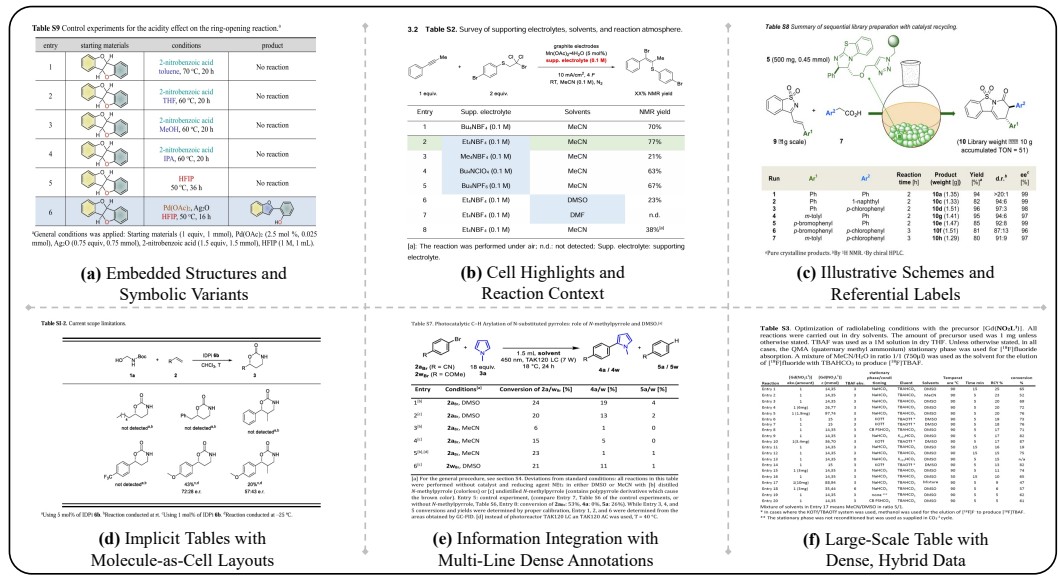

(a) Embedded Structures and Symbolic Variants

(b) Cell Highlights and Reaction Context

(c) Illustrative Schemes and Referential Labels

(d) Implicit Tables with Molecule-as-Cell Layouts

(e) Information Integration with Multi-Line Dense Annotations

(f) Large-Scale Table with Dense, Hybrid Data

Figure 1: Illustrative examples from the ChemTable dataset, showcasing the diverse and multimodal challenges for Multimodal Large Language Models.

understanding. This gap motivates our work: to build a comprehensive benchmark that captures the structural complexity, semantic richness, and reasoning demands of real-world chemical tables.

Among scientific tables, those found in chemistry pose especially complex challenges that go far beyond standard layout parsing (Abdelmagid et al., 2014). A typical chemical table encodes rich experimental workflows through dense symbolic notation (e.g., reagents, ligands), domain-specific abbreviations (e.g., "BINAP", "TFA"), and embedded visual elements such as molecular structures or reaction schemes (Tarasova et al., 2022; Leung et al., 2024). As illustrated in Figure 1, a single row often represents a multi-variable configuration—catalyst, ligand, solvent, additive, with quantitative outcomes like yield, selectivity. These tables also rely on implicit conventions (e.g., ratio formats like ">19/1") and footnoted exceptions, making their semantics both subtle and compact. **Interpreting such tables requires aligning symbolic, numeric, and visual information in a domain-aware manner**, which presents significant challenges for general-purpose MLLMs not trained on scientific representations (Li et al., 2024b; Abdelmagid et al., 2014).

To systematically address these challenges, we introduce **ChemTable**, a high-quality benchmark designed for recognition and understanding in chemical tables. The dataset comprises over **1,300 tables** curated from peer-reviewed chemistry literature, spanning diverse reaction types, experimental conditions, and reporting formats. ChemTable supports two core tasks: **table Recognition**(Zhang et al., 2025)—including table structure reconstruction and content extraction—and **table Under-standing** (Ruan et al., 2024; Cheng et al., 2025), formulated through more than **9,000 QA instances** across two categories: (1) *descriptive questions*, which evaluate a model's ability to extract key facts; and (2) *reasoning questions*, which require comparison, attribution, and domain-grounded inference.

To enable scalable and consistent evaluation, all answers follow a *short-form format* suitable for MLLMs-based automatic grading. We benchmark seven MLLMs for table recognition and ten MLLMs for table understanding, covering both open-source and proprietary families, and observe significant performance gaps across all tasks. Compared to human and expert performance, current MLLMs are behind—particularly in interpreting symbolic and embedded graphical elements. Our analysis further reveals key insights, including the symbolic understanding gap, the limited trans-ferability of MLLMs to scientific domains. We release ChemTable, along with evaluation tools, to support future research in multimodal scientific understanding.

## 2 RELATED WORK

### 2.1 EXISTING BENCHMARKS ON TABLE RECOGNITION.

Existing table recognition benchmarks mainly focus on two tasks: table structure recognition and table recognition. Early studies on table recognition relied on small but high-quality datasets such as

ICDAR-2013 (Göbel et al., 2013). Since 2019, large-scale datasets (Zhong et al., 2019; Gao et al., 2019; Li et al., 2020; Smock et al., 2022) have reshaped the field, enabling the deep learning era of table recognition. However, their annotations are programmatically generated and provide only table structures without cell content, limiting their utility in deeper understanding tasks. To enable deeper semantic understanding, recent datasets such as FinTabNet (Zheng et al., 2021), PubTabNet (Zhong et al., 2020), and SciTSR (Chi et al., 2019) incorporate logical cell locations and detailed content annotations, with TabRecSet (Yang et al., 2023) further extending coverage through multilingual and polygon-based labeling.

Beyond the general domain, scientific tables introduce richer layouts and domain-specific elements. Chemical tables, in particular, are uniquely challenging: they feature dense symbolic notation, embedded molecular structures, and implicit conventions, yet remain indispensable for reporting experimental knowledge. This complexity not only makes them valuable for practical applications but also an ideal testbed for probing the capabilities of multimodal large language models. Nevertheless, dedicated benchmarks for chemical table recognition are still lacking.

## 2.2 Existing Benchmarks on Table Understanding.

Several datasets have been proposed for table understanding tasks. Among them, WikiTQ (Pasupat & Liang, 2015) contains tables extracted from Wikipedia paired with natural language questions and has become a widely used early benchmark in this area. In recent years, advancements in natural language processing have extended beyond traditional homogeneous tables to incorporate additional modalities. For instance, HybridQA (Chen et al., 2020) and MMTab (Zheng et al., 2024) introduce multi-modal or semi-structured sources for more complex reasoning. Moreover, domain-specific benchmarks such as FinQA (Chen et al., 2021) (financial) and SciTab (Lu et al., 2023b) (scientific) address unique challenges by integrating structured tables with related textual content to support complex reasoning tasks.

The ChemTable dataset is different from existing benchmarks in several key ways. It is a comprehensive, open-ended dataset focused on table recognition and understanding in chemistry, created to support scientific question-answering systems. Unlike other datasets, its questions combine visual elements, table data, and chemistry knowledge, making it necessary for models to recognize table content and structure, then reason across that information. This setup reflects real tasks that researchers face when drawing conclusions from data.

Table 1: Comparison of table understanding datasets. We use the following shorthand: Dom. Spe. = Domain Specific, Pict. Moda. = Pictorial Modality, Text. Moda. = Textual Modality, Human Writ. = Human Written, LLM Gene. = LLM Generated.

| Name | Table Modal | | | Question Source | |
| | Dom. Spe. | Pict. Moda. | Text Moda. | Human Writ. | LLM Gene. |
| --- | --- | --- | --- | --- | --- |
| WikiTQ Pasupat & Liang (2015) | ✗ | ✗ | ✓ | ✓ | ✗ |
| WikiSQL Zhong et al. (2017) | ✗ | ✗ | ✓ | ✓ | ✗ |
| HybridQA Chen et al. (2020) | ✗ | ✗ | ✓ | ✓ | ✗ |
| FinQA Chen et al. (2021) | ✓ | ✗ | ✓ | ✓ | ✗ |
| SciTab Lu et al. (2023b) | ✓ | ✗ | ✓ | ✓ | ✓ |
| MMTab Zheng et al. (2024) | ✗ | ✓ | ✗ | ✓ | ✓ |
| **ChemTable** | ✓ | ✓ | ✓ | ✓ | ✓ |

# 3 ChemTable: A Chemical Table Recognition and Understanding Benchmark

We introduce ChemTable, a benchmark that systematically curates chemical research tables, evaluates table recognition capabilities, and advances table-based question answering in the chemical domain.

## 3.1 Dataset Construction and Annotation

In this section, we outline our methodology for systematically collecting and annotating chemical research tables to construct domain-specific datasets. We first describe the data sources and table types, followed by the introduction of a structured data annotation protocol that seamlessly integrates structural features (e.g., table structural information, text formatting) with chemical elements.

### 3.1.1 Table Collection

**Journal Selection.** To ensure both credibility and disciplinary relevance, we selected publications from top-tier chemistry journals (e.g., ACS Catalysis, JACS, Chem, Angewandte Chemie Int. Ed.,

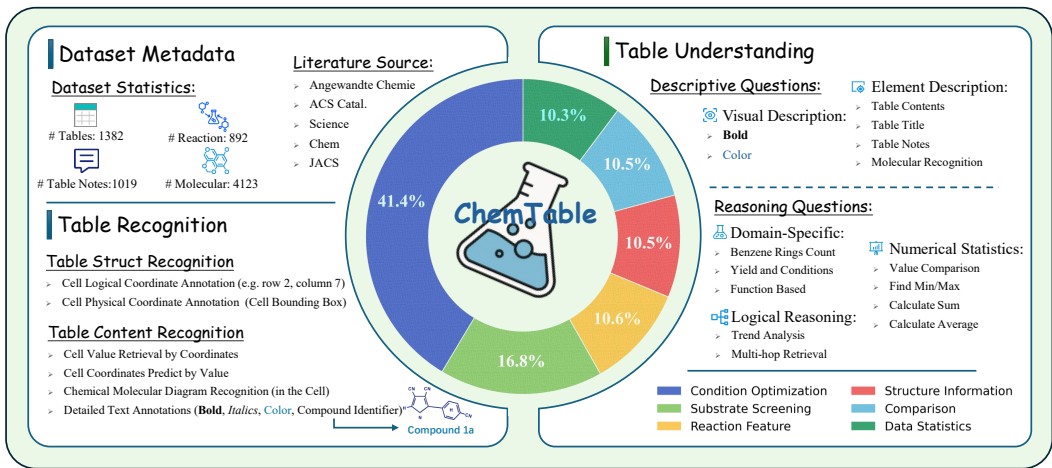

Figure 2: Overview of metadata, table recognition, and table understanding in ChemTable.

and Science). The dataset covers the past decade (2015–2024), providing a balanced scope that captures recent advances while retaining representative studies across the field. All publications were systematically linked to their DOIs to guarantee traceability and to facilitate future validation or extension of the dataset. Copyright considerations regarding data usage and redistribution are detailed in Appendix S.

**Table Categorization Strategy.** The dataset comprises six primary table types: (1) condition optimisation tables, (2) substrate screening tables, (3) chemical structure information tables, (4) reaction feature data tables, (5) property/result comparison tables, and (6) data statistics tables. Condition optimisation and substrate screening tables comprise over 50% of the dataset.

### 3.1.2 TABLE ANNOTATION

For table annotation, we formed a dedicated team to identify table titles, annotations, main content, and additional captions (Desai et al., 2021). Each element was annotated with pixel coordinates and OCR-verified text. We also encoded the logical structure of tables, such as row-column relationships. To preserve visual meaning, we carefully recorded stylistic features like boldface, italics, and color. For table-based question answering annotation, descriptive questions are directly derived from the detailed annotations of table elements created in the previous step. Simple reasoning questions are generated using GPT-4.1 (OpenAI, 2025) and then filtered based on difficulty. For more complex reasoning questions and visually descriptive tasks, we manually annotated 2,122 questions with the help of experienced graduate students. These questions focused on specific topics, such as reaction conditions and yields, while allowing a variety of question styles. Annotators were also encouraged to include unanswerable questions caused by missing data, vague references, or incomplete formatting. To ensure accuracy, all questions were verified by human review and model checks. Details can be found in the Appendix D.

### 3.2 TABLE RECOGNITION

Table recognition is a fundamental step in document understanding, which aims to turn table images into structured data. This task is more challenging in chemistry because of complex layouts and specialized symbols (Li et al., 2024a).

### 3.2.1 TASK DEFINITION

We employ the generation paradigm of MLLMs to address the table recognition (TR) task, which is formulated as a format mapping problem from images to sequences. Formally, given a dataset $\mathcal{D} = \{(I^i, H^i)\}_{i=1}^n$ with $n$ samples, we predict the corresponding structured form $H^i$ for each table image $I^i$. Specifically, we provide the image table $I^i$ along with a prompt $P$ as input to the MLLMs, which generates the structured data form $\hat{H}^i = \text{MLLM}(P, I^i)$.

### 3.2.2 EVALUATION PROTOCOLS

To further assess the capabilities of table recognition models and understand the challenges of chemical-domain data, we propose a set of tasks focused on domain-specific reasoning in chemical table understanding. Specifically, we introduce the following three tasks:

**Value Retrieval:** This task evaluates a model's ability to locate and extract cell-level content accurately. Given a table and a pair of coordinates, the model must return the exact value in that cell. This task directly measures the model's precision in structured data parsing and positional alignment.

**Position Retrieval:** This task requires the model to infer the position of a specific value. Given a table and a target value, the model must identify the correct row and column. This tests the model's understanding of value localisation within structured layouts.

**Molecular Recognition:** Chemical tables often include molecular structures represented as images, either embedded within cells or positioned externally. This task aims to evaluate a model's ability to recognise and interpret such molecular graphics. The objective is to extract the corresponding SMILES (Simplified Molecular Input Line Entry System) string from a molecular diagram. This task presents unique challenges, such as fine-grained visual understanding and domain-specific symbol interpretation, which are not typically encountered in general-domain table recognition (Morin et al., 2023; Han et al., 2024).

## 3.3 TABLE UNDERSTANDING

We divide table comprehension evaluation into two types: descriptive questions and reasoning questions. Descriptive questions test the model's ability to extract and summarize basic table information, while reasoning questions assess its ability to perform deeper analysis and inference. To improve the quality of the evaluation, we applied a data filtering process to increase both the diversity and difficulty of the questions in our dataset.

Table 2: ChemTable dataset statistics. unique tokens and QA lengths are calculated based on the Qwen2.5-7B tokenizer.

| Statistics | Value |
|---|---|
| **Images** | |
| Total Images | $1,382$ |
| Years | $2015 - 2024$ |
| Average size (px) | $3687 \times 4086$ |
| **Descriptive Questions** | |
| # questions | $7,344$ |
| # unique questions | $1,512$ |
| *Question* | |
| - # unique tokens | $1,568$ |
| - maximum length | $25$ |
| - average length | $11.10$ |
| *Answer* | |
| - # unique tokens | $12,032$ |
| - maximum length | $148$ |
| - average length | $8.99$ |
| **Reasoning Questions** | |
| # questions | $2,542$ |
| # unique questions | $1,735$ |
| *Question* | |
| - # unique tokens | $2,086$ |
| - maximum length | $37$ |
| - average length | $12.66$ |
| *Answer* | |
| - # unique tokens | $2,610$ |
| - maximum length | $78$ |
| - average length | $6.21$ |

### 3.3.1 TASK DEFINITION

We employ the generation paradigm of MLLMs to address the table question answering task. Formally, given samples, each sample consists of a table $T^i$, a natural language question $Q^i$, and the corresponding answer $A^i$. To answer the question, we provide the table $T^i$ and the question $Q^i$ as input to the MLLMs in the form of a prompt $P$, yielding the predicted answer. In the visual table QA setting, $T^i$ is an image of a table, while in the text table QA setting, $T^i$ is a structured textual representation.

### 3.3.2 DESCRIPTIVE QUESTIONS.

Descriptive questions aim to provide a general overview of the basic information presented in the table shown in the image. These questions include: 1) describing the main body of the table, such as its dimensions; 2) describing basic metadata of the table, including titles and notes; 3) describing domain-specific elements, such as reaction conditions in chemical reaction formulas or SMILES notation in molecular graphs; and 4) identifying certain visual features in the table, such as rows highlighted with special colors. Chemical tables often contain images, such as molecular structures, instead of plain text, making them harder to read and analyze. For example, an image in a table row can affect spacing and make it harder to recognize content. Sometimes, only certain parts of a molecule, like a red-highlighted –OH group—are shown, which adds to the difficulty for models trying to understand the table.

### 3.3.3 REASONING QUESTIONS

Reasoning questions are designed to further analyze and infer information from the data presented in the table within the image. They require a comprehensive understanding to make informed judgments. These questions include: 1) Numerical and statistical reasoning, 2) Trend and change analysis, 3)

Multi-hop logical reasoning, and 4) Domain-specific reasoning. Specific details are provided in the Appendix F.1. Since the data in chemical tables is often tightly linked to specific experimental conditions, molecular structures, and graphical annotations, reasoning questions not only assess the ability to comprehend explicit information but also place greater demands on domain knowledge, the integration of information across rows and columns, and the ability to reason using visual symbols.

### 3.3.4 DATA FILTERING

**Diversity.** We identified questions with overly repetitive structures and semantically similar phrasing to encourage a broad range of question formulations. These were rewritten using GPT-4.1 with prompt templates provided in the Appendix Q. We selected algorithms that maximised the semantic distance from the original question, enhancing the linguistic and structural diversity of the dataset.

**Difficulty.** We implemented a filtering strategy to ensure the dataset poses a meaningful challenge. We first conducted a single-pass QA evaluation using the Qwen-2.5-7B model (Yang et al., 2024). For each question, we recorded whether the model was able to produce the correct answer on the first attempt. Questions that were answered correctly in one try were deemed too simple. To filter out these low-difficulty samples, we randomly discarded some of them. This approach allowed us to enrich the dataset with more challenging examples that were better suited for evaluating advanced reasoning capabilities.

## 4 EXPERIMENTS ON TABLE RECOGNITION

### 4.1 EXPERIMENTAL SETUP

**Evaluation Metrics.** To evaluate the performance of the table recognition task, we adopt the improved similarity metric based on tree edit distance (TEDS) (Zhong et al., 2020) along with the TEDS-structure indicator. Specifically, the table content may contain molecular graphs in chemical scenarios. Interpreting these molecular graphs using the simplified molecular input line entry system(SMILES) can result in structural isomorphism, where different representations or atom orders may correspond to the same molecule, making it inappropriate for TEDS to utilize normalised edit distance as a measure of cell content recognition accuracy. Therefore, for cells containing chemical molecular graphs, we replace the normalised edit distance with the Tanimoto coefficient (Holliday et al., 1995) to accurately assess the performance of table recognition. We use accuracy (ACC) as the evaluation metric for fine-grained retrieval experiments.

**Baselines.** We evaluate a diverse set of MLLMs, including open-source models InternVL3-78B (Zhu et al., 2025), Llama-3.2-90B (Meta, 2024), and Qwen2.5-VL-72B (Bai et al., 2025), as well as proprietary models Gemini-2.5-Flash (Google, 2025), GPT-4.1, GPT-4.1-mini (OpenAI, 2025), and Claude-3.7-Sonnet (Anthropic, 2024). Implementation details are in Appendix L.

### 4.2 EXPERIMENTAL RESULTS

**Main Results Analysis.** We evaluate the table recognition performance using an improved similarity metric based on TEDS, along with the TEDS-structure indicator. The results are in Table 3 and key findings are listed as follows.

**(a) Small performance gap between open-source and proprietary models.** Although a performance gap between open-source and proprietary models still exists, both achieve promising results in table recognition. For example, Gemini-2.5-Flash achieves 95.91 on TEDS-Struct and 88.29 on TEDS, while the open-source Qwen2.5-VL also performs competitively, scoring 93.12 on TEDS-Struct and 89.45 on TEDS. This shows that both model types possess strong capabilities in table understanding and structure reconstruction.

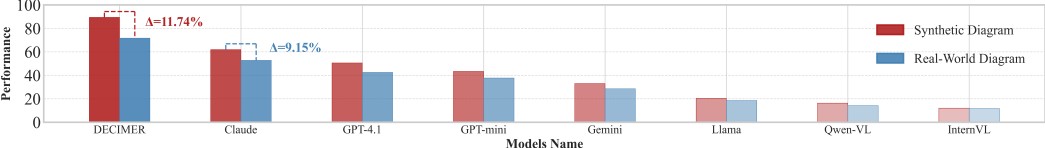

Figure 3: Comparative evaluation of MLLMs and DECIMER for molecular formula recognition from real-world and synthetic chemical diagrams.

Table 3: Performance of different MLLMs on table recognition and fine-grained retrieval tasks. TEDS ↑, TEDS-Struct ↑, and ACC↑ are used as the evaluation metric. The "*" indicates using the tanimoto coefficient for molecular formula prediction from molecular diagrams within table cells.

| Model Category | Model Name | Table Recognition | | Value Retrieval | Position Retrieval |
|---|---|---|---|---|---|
| | | TEDS-Struct* | TEDS* | ACC | ACC |
| Proprietary | Claude-3-7-Sonnet | 92.58 | 85.40 | **33.89** | **53.06** |
| | GPT-4.1 | 95.48 | **88.93** | 29.60 | 49.49 |
| | Gemini-2.5-Flash | **95.91** | 88.29 | 29.19 | 36.92 |
| | GPT-4.1-mini | 95.25 | 87.50 | 17.01 | 35.16 |
| Open-Source | Qwen2.5-VL | 93.12 | 89.45 | 31.72 | 38.35 |
| | InternVL3 | 94.40 | 86.06 | 29.58 | 33.91 |
| | Llama-3.2 | 93.15 | 87.46 | 29.30 | 32.85 |

**(b) Poor fine-grained retrieval across all MLLMs.** All models show weak performance on fine-grained retrieval tasks, such as locating cell content by row-column positions or inferring positions from content. As shown in Table 3, even Claude-3.7-Sonnet, the best-performing model, only achieves an accuracy of 33.89 on value retrieval, with others performing worse. This highlights ongoing challenges in achieving precise fine-grained alignment in current MLLMs.

**(c) Molecular formulas pose a key recognition bottleneck.** Recognition performance declines as the number of molecular formulas in tables increases. In contrast, accuracy improves significantly when molecular formulas are absent. Specifically, models perform notably worse on chemical tables with many molecular structures than on those with plain text or simple layouts. This indicates that molecular formulas remain a key bottleneck for current models and require further optimization to improve recognition accuracy.

**Analysis of Molecular Formula Recognition.**
We evaluated MLLMs on molecular formula recognition from real-world academic papers and synthetic diagrams. As shown in Figure 3, MLLMs can identify and convert molecular structures into chemical formulas, but their performance on real-world diagrams is lower than on synthetic ones, highlighting the impact of data diversity and quality on model robustness. Although MLLMs possess some chemical domain knowledge, their accuracy and reliability remain significantly inferior to specialized models (DECIMER(Rajan et al., 2023)). This gap underscores the need for further improvements to enable MLLMs to effectively handle domain-specific tasks in advanced scientific applications.

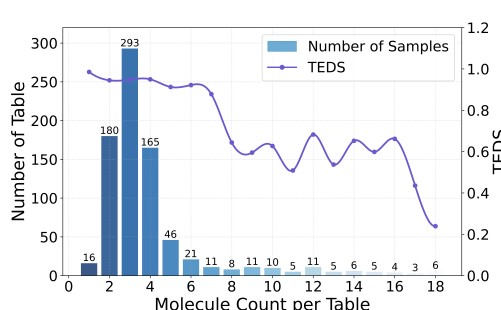

Figure 4: Impact of molecular formula complexity on table recognition.

**Analysis of Chemical Representation Recognition.** We examine whether current MLLMs treat chemical representations (e.g., molecular formulas and structural diagrams) as noise when processing tables, as shown in Figure 4. Results indicate that even when molecular structures appear only in the surrounding context—not within the table itself—their presence still degrades performance. This suggests that chemical symbols can interfere with the model's overall understanding. Our findings confirm that MLLMs struggle to parse and integrate multimodal chemical information, revealing a key limitation in chemical table understanding and an important direction for future improvement.

## 5 EXPERIMENTS ON TABLE UNDERSTANDING

### 5.1 EXPERIMENTAL SETUP

**Evaluation Metrics.** For tasks requiring descriptive answers, we use edit distance (Levenshtein et al., 1966) as the primary metric to assess recognition accuracy. For more open-ended question answering (QA) tasks, we adopt a binary evaluation strategy powered by GPT-4.1-nano (OpenAI,

Table 4: Performance comparison of MLLMs on descriptive and reasoning tasks with human in chemical table understanding. Overall, MLLMs perform impressively but are slightly outperformed by humans in complex tasks. We denote the best score in blue , and the second-best score in green .

| Question Type | | GPT-5 | Gemini-Pro | Claude-4.5 | GPT-4.1 | Gemini | Claude-3.7 | GPT-mini | Qwen-VL | InternVL | Llama | Human |
|---|---|---|---|---|---|---|---|---|---|---|---|---|
| *Descriptive Questions* | | | | | | | | | | | | |
| Element Description | Table Dimensions | 74.89 | 74.35 | **76.11** | 73.07 | 73.10 | 75.39 | 68.31 | 71.42 | 70.50 | 70.27 | - |
| | Title Description | 83.74 | **87.67** | 87.30 | 87.31 | 81.64 | 81.03 | 84.79 | 83.18 | 84.35 | 85.50 | - |
| | Annotation Description | **93.11** | 89.91 | 87.41 | 92.94 | 81.12 | 68.93 | 81.86 | 90.23 | 73.27 | 81.39 | - |
| | Molecular Recognition | 52.04 | **69.31** | 58.14 | 42.49 | 28.47 | 52.71 | 37.62 | 14.14 | 11.63 | 18.50 | - |
| Visual Description | Bold Description | 40.53 | 45.81 | 48.93 | 44.27 | 41.22 | 50.38 | 35.88 | 38.93 | 32.82 | 52.73 | **98.99** |
| | Color Description | 50.78 | 54.56 | 48.19 | 56.48 | 41.22 | 50.38 | 54.92 | 58.55 | 49.74 | 58.03 | **97.73** |
| *Reasoning Questions* | | | | | | | | | | | | |
| Domain-Specific QA | Benzene Rings Count | 57.22 | 63.67 | 46.00 | 52.31 | 75.32 | 62.83 | 49.51 | 59.61 | 47.66 | 21.97 | **94.98** |
| | Yield and Conditions | 90.53 | 92.69 | 90.81 | 89.14 | 90.97 | 89.42 | 89.97 | 85.24 | 74.93 | 82.13 | **93.61** |
| | Function Based | 37.94 | 73.97 | 45.66 | 37.30 | 71.70 | 62.06 | 20.78 | 35.37 | 30.23 | 25.83 | **89.27** |
| Numerical Statistics | Value Comparison | 86.44 | 92.00 | 91.85 | 91.80 | 92.00 | 93.60 | 78.40 | 94.40 | 67.14 | 81.45 | **100.00** |
| | Find Min/Max | 86.18 | 89.62 | 94.79 | 85.85 | 83.18 | 94.39 | 79.44 | 94.39 | 60.94 | 60.95 | **100.00** |
| | Calculate Sum | 60.65 | 56.43 | 46.32 | 58.33 | 46.43 | 53.68 | 32.63 | 32.63 | 24.07 | 34.12 | **100.00** |
| | Calculate Average | 47.84 | 55.00 | 50.85 | 44.87 | 46.82 | 46.75 | 22.52 | 26.13 | 24.19 | 33.33 | **98.20** |
| Logical Reasoning | Trend Analysis | 84.46 | 87.32 | 86.21 | 81.87 | 86.53 | 83.94 | 75.13 | 74.61 | 76.34 | 55.96 | **98.45** |
| | Multi-hop Retrieval | 83.68 | 84.87 | 85.65 | 84.87 | 87.94 | 88.16 | 83.55 | 82.89 | 80.20 | 81.48 | **98.67** |

2025), which classifies each response as either correct or incorrect (Lu et al., 2023a; Young et al., 2024; Dubois et al., 2023),. The overall performance is then quantified by computing the accuracy (ACC) based on these binary classifications.

To ensure the reliability of this automated evaluation process, we randomly sampled 20% of the QA instances for manual verification. Human annotators reviewed the model outputs against reference answers and provided independent judgments. We then calculated the agreement rate between human evaluations and the binary assessments produced by GPT-4.1-nano . Implementation details are provided in the Appendix G.

**Baselines.** For table understanding, we evaluate a superset of the open-source and proprietary MLLMs introduced in Section 4.1. Concretely, we consider GPT-5, Gemini-2.5-Pro, Claude-4.5-Sonnet, GPT-4.1, Gemini-2.5-Flash, Claude-3.7-Sonnet, GPT-4.1-mini, Qwen2.5-VL-72B, InternVL3-78B, and Llama-3.2-90B. For brevity, we denote them as GPT-5, Gemini-Pro, Claude-4.5, GPT-4.1, Gemini, Claude-3.7, GPT-mini, Qwen-VL, InternVL, and Llama. We additionally include a human performance baseline (Human), consisting of human answers on the evaluation data (the collection process and evaluation protocol are detailed in Appendix L). We additionally report results for domain-specific chemistry models in Appendix K.

## 5.2 EXPERIMENTAL RESULTS.

We compare a set of representative MLLMs on the table understanding task in Table 4, covering both descriptive and reasoning questions. Below are our main findings:

**(a) General reasoning strength, numerical weakness.** Across general reasoning tasks such as trend analysis, value comparison, and finding min/max values, MLLMs achieve relatively high accuracy. For example, Gemini-Pro reaches 87.32 ACC on *Trend Analysis*, indicating that models can reliably capture basic quantitative and monotonic patterns from tables. However, performance drops sharply on arithmetic-heavy tasks: on *Calculate Average*, Gemini-Pro only attains 55.00 ACC, far below human performance, highlighting limitations in calculation-intensive reasoning.

**(b) Descriptive tasks outperform domain-specific chemistry QA.** MLLMs perform strongly on descriptive questions about table content. On *Annotation Description*, GPT-5 achieves 93.11 ACC, reflecting robust capabilities in recognizing and summarizing textual annotations. In contrast, accuracy decreases on domain-specific chemistry questions. Even the best model on *Function Based* QA, Gemini-Pro, reaches only 73.97 ACC, substantially below the human baseline, highlighting the difficulty of integrating chemical knowledge with visual table structure.

**(c) Visual style interpretation remains challenging.** Tasks that depend on visual or stylistic cues—such as boldface and color highlighting—remain particularly challenging. While Llama and Qwen-VL emerge as the top contenders in this category, their performance is far from ideal. Llama reaches 52.73 ACC on Bold Description, and Qwen-VL leads slightly with 58.55 on Color

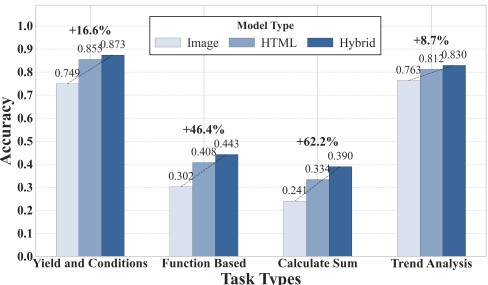

Figure 5: Accuracy comparison of InternVL3-78B on chemistry table understanding tasks across input modalities

| Model | Missing | | Ambiguity |
| --- | --- | --- | --- |
| | Col/Row | Style | |
| Llama | 66.43 | 77.31 | 80.08 |
| GPT-mini | 70.29 | 35.64 | 70.36 |
| InternVL | 74.84 | 46.15 | 62.80 |
| Qwen-VL | 75.51 | 76.27 | 97.64 |
| GPT-4.1 | 84.02 | 81.21 | 79.69 |
| Gemini | 92.50 | 73.06 | 93.67 |
| Claude-3.7 | **97.12** | **90.68** | **98.23** |

Table 5: Model performance on unanswerable question categories.

Description. However, compared to the near-perfect human performance (>97 ACC), this substantial gap suggests that fine-grained visual formatting is still poorly grounded in current MLLMs.

**(d) Closed-source models dominate complex and domain tasks, but humans still lead.** Overall, closed-source models such as GPT-5, Gemini-Pro, Claude-4.5, and GPT-4.1 dominate complex reasoning and chemistry-specific tasks. For instance, Gemini-Pro achieves 73.97 ACC on *Function Based* QA and over 90 ACC on *Yield and Conditions*, whereas most open-source models lag behind on these tasks. At the same time, strong open-source models like Qwen-VL can match or surpass proprietary ones on certain numerical subtasks (e.g., *Value Comparison* at 94.40 ACC). Nevertheless, humans still outperform all models on the most complex table understanding tasks, indicating a non-trivial gap to expert-level competence.

## 5.3 Analysis of Multimodal Input and Model Behavior

**Impact of Input Modality.** In Figure 5, we evaluated three input modalities—Text QA (HTML), VQA (Image), and Hybrid QA (Hybrid)—to assess how different formats affect model performance in answering chemistry-related questions. Experimental results across tasks such as Yield and Conditions, Function Based show that Hybrid QA achieves the highest accuracy by combining textual and visual inputs, enabling a more comprehensive understanding of complex chemical structures. Text QA outperforms VQA, as converting images to text improves interpretability, although it may introduce errors due to information loss. In contrast, VQA struggles with detailed visual content, leading to higher error rates. These findings suggest that hybrid input strategies are most effective for enhancing performance, while careful handling of text conversion remains essential.

**Impact of Unanswerable Questions on Model Behavior.** We examine how advanced MLLMs handle unanswerable questions by refraining from responding, as shown in Table 5. This occurs when questions exceed model capabilities or lack context, which we classify into three types: non-existent content, missing format/style, and ambiguity. Our results show that leading models can effectively determine when not to answer by using contextual understanding and reasoning, reflecting a form of self-awareness that avoids misinformation. In contrast, smaller models often fail to recognize such cases, producing incorrect or irrelevant answers. This underscores the importance of model scale and training quality for reliable and trustworthy MLLMs in question-answering and prompt engineering.

## 6 Conclusion

In this work, we introduced **ChemTable**, a large-scale dataset and benchmark designed to evaluate multimodal large language models (MLLMs) on recognition and understanding tasks involving chemical tables. By curating over 1,300 real-world chemistry tables and annotating them with domain-specific metadata and question-answering tasks, ChemTable captured the multimodal, symbolic, and semantic challenges unique to the field. Our comprehensive evaluation revealed significant performance gaps between current MLLMs and human-level capabilities, particularly in domain-specific reasoning and molecular recognition. We believe that this dataset and benchmark will facilitate future advancements in multimodal scientific analysis and understanding.

**Ethics statement.** We confirm that this work aligns with accepted ethical standards in machine learning research. All datasets used in this study are derived from publicly available sources, and we carefully respect copyright and licensing requirements. Annotations were performed by qualified domain experts under fair working conditions. No personally identifiable or sensitive data were collected or used.

**Reproducibility statement.** To support reproducibility, we provide detailed descriptions of the dataset construction process, annotation protocols, and experimental setups, including model configurations, hyperparameters, and evaluation metrics, in the main text and appendices. We also release the benchmark, scripts, and evaluation tools to facilitate replication and further research.

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

APPENDIX

# A  FINE-GRAINED QA BEHAVIOR ANALYSIS

This section presents detailed analyses of how reasoning complexity and query directionality affect the accuracy and robustness of multimodal models in scientific table question answering.

## A.1  EFFECT OF QUESTION HOPS ON ANSWER ACCURACY

To understand the impact of reasoning complexity on model performance, we conduct a fine-grained analysis of multi-hop question answering, using hop counts of 2, 3, and 4 as indicators of increasing logical depth. As shown in Table 6, all models exhibit a monotonic decline in performance with increasing hop counts. For instance, Claude drops from 91.29% on 2-hop questions to 70.70% on 4-hop, while InternVL declines more sharply from 83.58% to 59.47%. This trend reflects a consistent rise in difficulty as models are required to perform more compositional and contextual reasoning over tabular data.

In the context of ChemTable, where tables encode dense, multimodal chemical knowledge—including symbolic notations, visual molecular structures, and complex reaction dependencies—multi-hop questions often require the integration of spatial, textual, and domain-specific knowledge. For example, answering a 4-hop question might involve comparing reaction conditions across multiple rows and applying chemistry knowledge such as identifying functional groups or evaluating yields under varying catalysts.

Table 6: Performance of Multimodal Models on Multi-Hop QA Tasks Categorized by Hop Count.

| Model | Hop Count | | | Overall |
|---|---|---|---|---|
| | 2 | 3 | 4 | |
| Claude | **91.29** | 84.98 | 70.70 | **88.16** |
| Gemini | 90.86 | **86.98** | 69.14 | 87.94 |
| GPT-4.1 | 87.72 | 79.71 | **71.78** | 84.87 |
| GPT-mini | 87.68 | 78.43 | 61.65 | 83.55 |
| Qwen-VL | 87.29 | 82.81 | 52.84 | 82.89 |
| Llama-3.2 | 84.59 | 80.18 | 61.78 | 81.48 |
| InternVL | 83.58 | 78.24 | 59.47 | 80.20 |

Our results show that stronger models such as Claude and Gemini maintain significantly higher accuracy on high-hop questions compared to smaller or open-source models (e.g., Qwen-VL or InternVL). This growing performance gap at higher hop levels suggests that complex multi-step reasoning tasks amplify the differences in model capabilities.

## A.2  ASYMMETRY IN CONDITION-YIELD TABLE REASONING

Directionality of query plays a significant role in multimodal table question answering. In this additional experiment, we evaluate four state-of-the-art multimodal large language models (MLLMs) on two complementary QA settings: (1) Forward Prediction, where the model predicts reaction outcomes (e.g., yield) based on given conditions; and (2) Inverse Prediction, where the model must infer conditions from known outcomes. As shown in Figure 6, all evaluated models demonstrate higher accuracy on the forward task compared to the inverse one. For instance, GPT-4.1 achieves 91.74% on forward prediction but drops to 86.82% on the

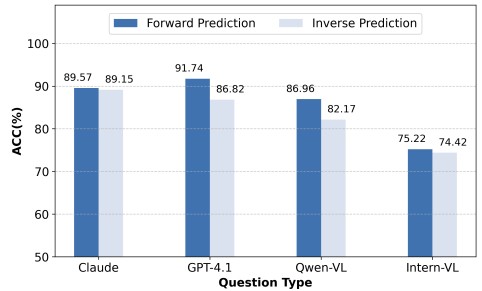

Figure 6: Accuracy of Multimodal Models in Answering Questions Given Conditions vs. Inferring Conditions from Outcomes.

inverse. This performance gap suggests that MLLMs are better aligned with natural scientific reasoning—where outcomes are typically deduced from conditions—than with the reverse logic. It also reflects an asymmetry in learned representations: while models can synthesize output from structured inputs effectively, they struggle more when tasked with deducing structured inputs from outcomes, which often requires multi-hop or abductive reasoning. These findings reveal a fundamental challenge for scientific understanding in reverse reasoning settings and highlight the need for targeted training strategies to enhance backward inference in MLLMs.

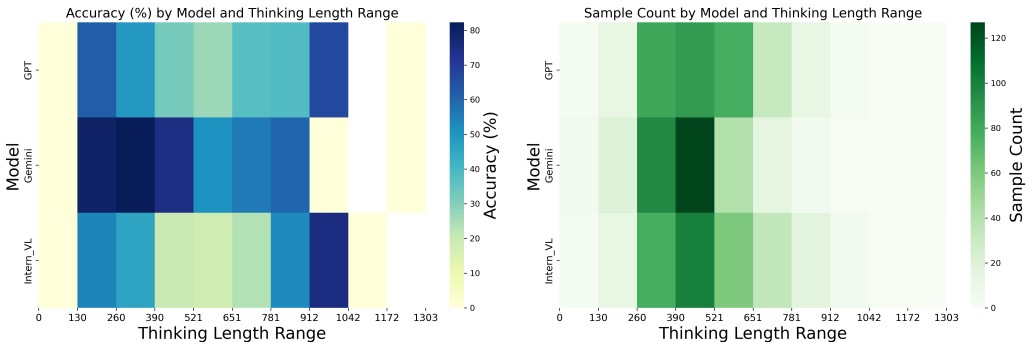

Figure 7: Accuracy and Sample Distribution by Thinking Length Across Models.

## B  CORRELATION BETWEEN RESPONSE LENGTH AND CORRECTNESS

To investigate how reasoning depth affects model performance, we analyzed the relationship between response (or "thinking") length and accuracy on Function-Based QA tasks. We evaluated three representative Multimodal Large Language Models—GPT-4.1, Gemini-2.5-Flash, and InternVL—by binning their outputs based on token length and computing corresponding accuracies and sample counts (Figure 7). This setting helps reveal whether longer responses lead to more accurate reasoning. As shown in the results, accuracy does not monotonically improve with longer thinking length. Instead, models tend to achieve peak performance at moderate length ranges. Beyond these ranges, accuracy either plateaus or decreases, likely due to verbosity, sample counts drop sharply at extreme lengths, limiting statistical confidence in those bins. Overall, the results suggest that effective reasoning often corresponds to an optimal response length—neither too short nor excessively long.

## C  EFFECT OF CHAIN-OF-THOUGHT REASONING

We conducted an ablation study using GPT-4.1 to evaluate the impact of Chain-of-Thought (CoT) prompting on multimodal question answering over chemical tables. Specifically, we compared model performance with and without CoT reasoning across four representative question types. As illustrated in Figure 8, the removal of CoT resulted in a consistent drop in accuracy, particularly for reasoning-oriented tasks. While annotation description—largely descriptive in nature—showed minimal change (92.94% with CoT vs. 92.35% without), substantial declines were observed in function-based questions (37.30% → 24.64%), summation (58.33% → 42.18%), and trend analysis (81.87%

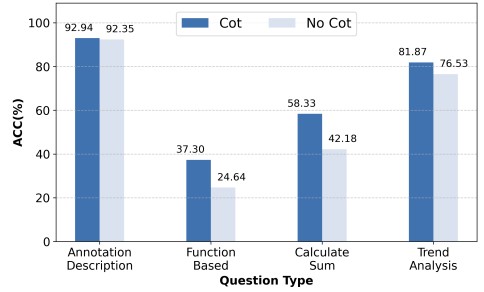

Figure 8: Impact of Chain-of-Thought (CoT) Reasoning on Question Answering Accuracy Across Different Task Types.

→ 42.18%), and trend analysis (81.87% → 76.53%). These results highlight the critical role of CoT prompting in enhancing the model's ability to perform step-by-step reasoning and complex inference. Overall, the findings underscore that even high-capacity models like GPT-4.1 benefit significantly from structured reasoning guidance, particularly in reasoning QA scenarios.

## D  SPECIFICATIONS OF ANNOTATION FORMAT AND PROCEDURE FOR TR

### D.1  ANNOTATION FORMAT

During the table image annotation phase, we conducted detailed annotations of chemical reaction tables to facilitate the structured extraction and downstream machine learning tasks. Specifically, we divided each chemical table into five distinct components: **Title**, **Reactions**, **Substances**, **Table**, and **Annotations**. The structure of each component is systematically defined to ensure consistency and interpretability across the dataset, as illustrated in Figure 9.

On the **left side** of Figure 9, we present a typical chemical reaction table that has been comprehensively annotated. This includes:

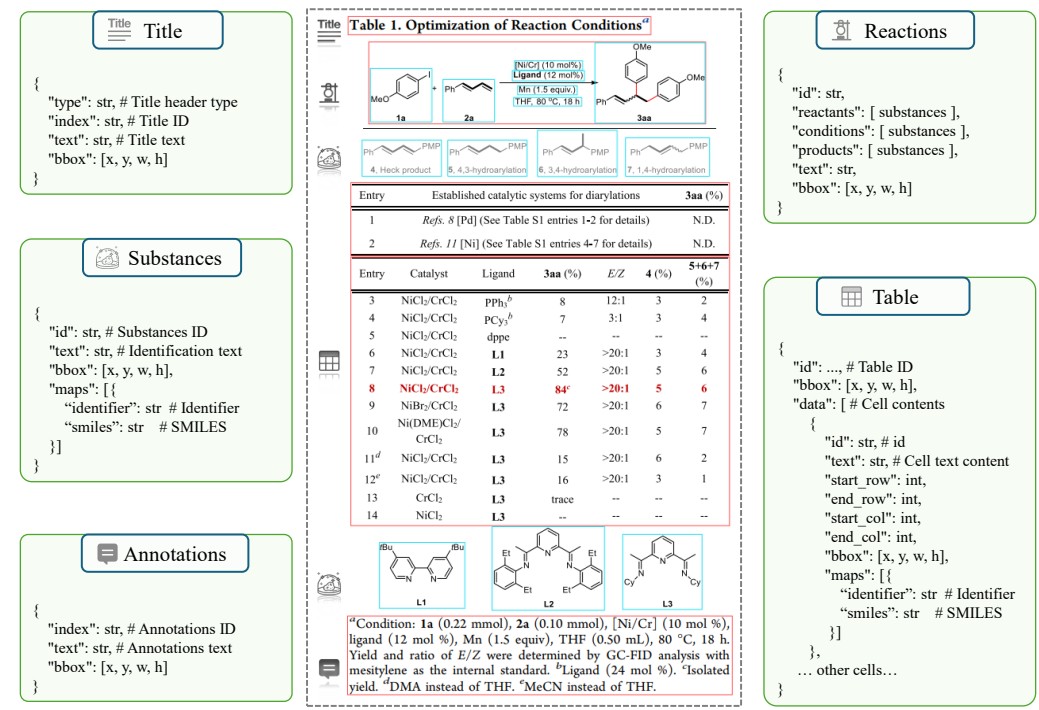

Figure 9: Structured Annotations of a Chemical Table into Title, Reactions, Substances, Table, and Annotations Components.

- **Title**: The title of the table, annotated with a bounding box, unique index, and its content. Indicate the subject or purpose of the table, such as the optimization of reaction conditions.

- **Reactions**: Schematic representations of chemical reactions including clearly separated reactants, products, and conditions. Each reaction entry is annotated with a unique identifier, text content, and a bounding box. Additionally, the involved chemical entities are linked to corresponding substance entries.

- **Substances**: All unique chemical structures in the table are annotated with bounding boxes and identification text. Each structure is mapped to its SMILES representation and identifier, ensuring its traceability and enabling computational applications.

- **Table**: The core tabular data, where each cell is annotated not only with a bounding box and textual content but also with logical coordinates indicating its specific row and column position in the table. Furthermore, chemical substances mentioned within the table cells are linked to the corresponding SMILES representations via a mapping mechanism.

- **Annotations**: Footnotes or explanatory text found below or around the table are included as annotations. These provide essential context such as experimental conditions or special notes on reagents and are annotated with bounding boxes and textual content.

On the **right side** of Figure 9, the data structure for each annotation component is illustrated using JSON-style schema definitions. This schematic defines the internal representation of each component in the dataset, including keys like `"bbox"`, `"text"`, `"maps"`, `"start_row"`, and `"start_col"`, enabling precise spatial and logical referencing within the table.

This annotation scheme ensures that both the visual layout and the semantic structure of chemical tables are faithfully captured, which is crucial for downstream applications such as automated chemical information extraction and chemical literature understanding.

## D.2 FINE-GRAINED TEXT ANNOTATION RULES

To further enhance the interpretability and structured utility of the dataset, we adopted a standardized markup protocol for annotating textual elements within the table components:

- **Reference Markers:** Textual references are annotated using the `\refmark{X}` syntax, where `X` is a unique identifier pointing to an associated explanatory note. Corresponding footnotes or commentary annotations are marked using `\mark{X}` within the **Annotations** component.

- **Substance Identifiers:** When referring to specific chemical structures or substances, the annotation uses `\refiden{X}` to denote in-text references, which link to formal definitions annotated via `\iden{X}` in the **Substances** component. This bidirectional referencing ensures clarity and consistency across the dataset.

- **Subscripts and Superscripts:** Chemical text annotations follow the convention of using the caret symbol (`^`) for superscripts and the underscore (`_`) for subscripts.

- **Text Formatting:** For stylistic features embedded in the body of the table (excluding titles and substance labels), the following LaTeX-style conventions are used:
  - Bold text: `\textbf{X}`. When the bolded text corresponds to a substance identifier, it is nested as `\textbf{\iden{X}}` or `\textbf{\refiden{X}}`.
  - Italicized text: `\textit{X}`.
  - Colored text: `\color{red}{X}`, where the color annotation reflects the visual emphasis found in the original table.

These conventions are designed to faithfully capture the nuanced visual and semantic cues present in scientific tables, which are critical for tasks involving automatic parsing, entity recognition, and domain-specific layout understanding.

### D.3 ANNOTATION PROCEDURE

The annotation process was carried out in three well-defined phases to ensure both coverage and precision across diverse types of chemical reaction tables.

**Phase I: Data Collection and Categorization.** We began by collecting and curating table images from peer-reviewed chemical literature published over the past decade. Specifically, we sourced documents from high-impact journals such as *ACS Catalysis*, *Journal of the American Chemical Society*, *Chem*, *Angewandte Chemie International Edition*, and *Angewandte Chemie*. From these publications, we systematically extracted regions explicitly labeled as tables based on captions, figure titles, or context cues. Following extraction, each table image was manually categorized into one of six functional types based on its primary purpose: (1) Reaction Condition Optimization Tables, (2) Substrate Scope Tables, (3) Chemical Structure Information Tables, (4) Reaction Feature Data Tables, (5) Property/Outcome Comparison Tables, and (6) Statistical Summary Tables. Among these, optimization and substrate scope tables comprised the majority (over 50%), while the remaining types each accounted for more than 10% of the dataset.

**Phase II: Coarse-Grained Annotation.** In the second phase, we performed coarse-grained annotations to establish the structural foundation of each table image. This included identifying and labeling five primary components — **Title**, **Reactions**, **Substances**, **Table**, and **Annotations**. For each component, bounding boxes and textual transcriptions were annotated. For instance, the **Title** region was delineated and transcribed to capture the overarching context of the table. Reaction schemes were segmented and annotated as **Reactions**, while molecular structures embedded in the table were marked as **Substances**. The main data matrix was annotated under the **Table** component, and any surrounding descriptive notes or footnotes were included under **Annotations**. This phase focused on accurately demarcating high-level semantic units to support later fine-grained processing.

**Phase III: Fine-Grained Annotation.** In the final phase, we applied fine-grained annotations to the core tabular content and reaction schematics. For the **Table** component, we annotated each individual cell with its bounding box, textual content, and logical position (row and column indices). If a cell referenced chemical entities, we linked the corresponding text or image to its associated SMILES identifier using a predefined mapping. Similarly, in the **Reactions** component, we annotated and linked specific chemical species — such as reactants, products, catalysts, or solvents — and reaction conditions (e.g., temperature, time, yield) to structured representations, enabling both visual and semantic disambiguation.

Following this three-stage annotation pipeline, we constructed a high-quality dataset comprising 1,500 fully annotated chemical table images. This dataset preserves both the visual layout and the underlying chemical semantics, laying a robust foundation for downstream tasks including machine learning model training, automated reaction information extraction, and chemical table understanding.

## E    ALGORITHM FOR CONVERTING ANNOTATIONS TO HTML

Since we only give the logical coordinate annotations of the tables, they cannot be directly used to calculate the TEDS metrics. To address this, we use the following pseudocode to convert the logical structure into markup sequence format. Firstly, we divide the entire conversion process into two stages. As shown in Algorithm 1, the first stage involves a preliminary preprocessing of the logical location information, associating the logical positions with the corresponding cell content and storing them in a tabular data matrix.

---

**Algorithm 1** From Logical Location to Tabular Data Matrix

---

**Input:** cells = { $C_1, C_2 \ldots, C_K$ }
**Output:** table

1:  max_row ← maximum value of 'end_row' in cells
2:  max_col ← maximum value of 'end_col' in cells
3:  Initialize table as a array with dimensions (max_row, max_col)
4:  **for** cell **in** cells **do**
5:      start_row, end_row, start_col, end_col, content ← cell
6:      rowspan ← $1 + $ end_row $-$ start_row
7:      colspan ← $1 + $ end_col $-$ start_col
8:      **for** row = start_row **to** end_row **do**
9:          **for** col ← start_col **to** end_col **do**
10:             **if** row == start_row **and** col == start_col **then**
11:                 table[row][col] ← { "rowspan": rowspan,
12:                 "colspan": colspan, "content": content }
13:             **else**
14:                 table[row][col] ← "merged"
15:             **end if**
16:         **end for**
17:     **end for**
18: **end for**

---

In the second stage, we traverse the tabular data matrix row by row, gradually converting the stored logical information and cell content into a mark-up sequence, eventually generating the conversion result. The specific implementation is detailed in Algorithm 2. The corresponding source code is available in our GitHub repository.

---

**Algorithm 2** From Tabular Data Matrix to Markup Sequence

---

**Input:** table
**Output:** markup

1:  Initialize markup ← "`<table>`"
2:  **for** row **in** table **do**
3:      markup += "`<tr>`"
4:      **for** cell **in** row **do**
5:          **if** cell == "merged" **then**
6:              **continue**
7:          **else**
8:              rsp = cell["rowspan"]
9:              csp = cell["colspan"]
10:             markup += `<td rowspan = ` $rsp$ ` colspan = ` $csp$ `>`
11:             markup += cell["content"] + "`</td>`"
12:         **end if**
13:     **end for**
14:     markup += "`</tr>`"
15: **end for**
16: markup += "`</table>`"

---

## F  WORKFLOW FOR QUESTION ANNOTATION

The process of question annotation in ChemTable is structured into four complementary stages: rule-based automatic generation, LLM-assisted synthesis, manual refinement, and domain-specific function-based annotation. Together, these stages ensure that the resulting question-answer pairs are both scalable and chemically meaningful. Each stage contributes a distinct layer of complexity and reasoning depth—from surface-level descriptions to functionally grounded scientific inquiry—allowing ChemTable to comprehensively cover the diverse landscape of tabular chemical data.

### F.1  RULE-BASED AUTOMATIC ANNOTATION

The first stage of question annotation is grounded in the structured annotations obtained from the preceding Table Recognition phase. Specifically, we utilize layout and semantic information such as table title, annotation blocks, cell types, and molecular elements. A set of deterministic scripts are designed to automatically generate descriptive question-answer (QA) pairs based on predefined heuristics. These rules capture basic factual and metadata-related queries, such as:

- Table dimensions (e.g., number of rows and columns),
- Title description (extracting the title),
- Annotation interpretation (e.g., footnotes or notes),
- Molecular recognition (e.g., identifying the molecular structure type in a cell).

This rule-based method ensures high coverage and consistency across common question types, especially those targeting descriptive understanding without the need for deep inference.

### F.2  LLM-ASSISTED GENERATION FOR SIMPLE REASONING

For numerical and statistically oriented reasoning tasks, we adopt a semi-automatic pipeline leveraging large language models (LLMs), specifically GPT-4.1. The question generation follows a prompt-based paradigm where we input the HTML representation of the table, the table image, and a specified reasoning type (e.g., comparison, summation) into a structured prompt template (see Section Q for details). The model then outputs a set of QA pairs. The LLM is guided to focus on quantifiable patterns and basic statistical operations, such as:

- Value Comparison (e.g., comparing yields across rows),
- Find Min/Max (e.g., identifying the entry with the highest selectivity),
- Calculate Sum (e.g., summing up yields in a column),
- Calculate Average (e.g., computing the mean conversion).

### F.3  MANUAL ANNOTATION FOR COMPLEX REASONING

For questions involving complex domain-specific logic or requiring visual-semantic integration—such as multi-hop reasoning, ambiguous references, or molecular structure interpretation—we rely on manual annotation. Annotators use an internal tool, LabelStudio (see Section H), where they are presented with the image of the table, metadata, and a set of predefined question types.

Human annotators are instructed to create diverse and challenging questions that demand:

- Domain knowledge (e.g., understanding catalyst-function relationships),
- Logical inference (e.g., combining footnotes with table entries),
- Visual decoding (e.g., counting specific molecular motifs like benzene rings).

To ensure quality, each question undergoes two rounds of review: first by MLLMs validation and then through another annotator validation. Annotators are also encouraged to include unanswerable questions–caused by non-existent content, missing format/style, and ambiguity—to reflect real-world data imperfections to further test model robustness.

### F.4 Domain-Specific Question Annotation: Function-Based QA

While prior work in scientific table QA has largely focused on fixed question templates—such as yield estimation, comparison, or description—chemical tables exhibit significantly broader functional diversity. In ChemTable, we identify a substantial subset of tables whose structure and purpose deviate from conventional paradigms. These include, but are not limited to, substrate screening matrices, catalyst performance evaluations, structure-property tables, and experimental condition explorations. Unlike standard output-driven tables (e.g., yield-focused), these tables encode specific scientific functions that are often implicit and domain-specific.

To address this, we introduce a novel annotation paradigm termed *Function-Based QA*, which aims to capture questions grounded in the functional roles of tables within experimental workflows. This process is semi-automated and comprises the following pipeline:

1. **Function Summary Generation.** We begin by prompting GPT-4.1 with the full HTML representation and image of a table, guiding it to generate a concise natural language summary that articulates the table's experimental function, purpose, or analytical focus.

2. **Function-Aligned Question Generation.** Based on the generated summary and table contents, we prompt GPT-4.1 and Claude to produce candidate QA pairs that probe aspects closely tied to the table's described function. These include nuanced inquiries such as the effect of a specific ligand under a fixed condition, rationale for substrate ordering, or interpretation of experimental design variables.

3. **Validation via Multi-Round QA.** To verify correctness and answerability, each candidate question undergoes three rounds of independent answering by GPT-4.1 and Claude-3.7-Sonnet. If all answers are consistent and correct across rounds, the question is accepted as valid. If discrepancies arise, human annotators inspect the question-answer pair for correctness and revise or discard as needed.

This annotation strategy enables ChemTable to extend beyond rigid QA formats, capturing richer scientific inquiry styles that reflect how chemists interpret and utilize tabular data. The resulting Function-Based QA subset significantly improves the coverage of the benchmark on real-world analytical reasoning.

## G Consistency Analysis: Verification of Human vs. MLLM

To validate the reliability of our automatic QA evaluation pipeline powered by GPT-4.1-nano, we randomly sampled 20% of the QA instances for manual verification. Two human annotators with chemistry backgrounds independently judged whether model-generated answers were correct, referencing the original table content and gold answers. We calculated agreement accuracy between human and automated judgments using simple percentage overlap:

$$\text{Agreement Rate} = \frac{|\mathcal{J}_{\text{human}} \cap \mathcal{J}_{\text{GPT}}|}{|\mathcal{J}_{\text{sampled}}|}, \tag{1}$$

where $\mathcal{J}_{\text{human}}$ and $\mathcal{J}_{\text{GPT}}$ denote the sets of instances labeled as correct by human annotators and GPT-4.1-nano, respectively, and $\mathcal{J}_{\text{sampled}}$ is the total set of sampled QA instances used for verification. The comparison between human annotations and GPT-4.1-nano's binary classifications showed a high agreement rate of 96.8% overall. Agreement was particularly strong for descriptive and numerical tasks. These results confirm that GPT-4.1-nano provides a reliable and scalable approximation of human judgment for most evaluation scenarios in ChemTable.

## H Screenshots of the Annotation Interface

We utilize LabelStudio as the primary platform for all data annotation tasks. The system is deployed on our internal computing clusters, allowing annotators to securely access the annotation interface via SSH forwarding. As shown in Figure 10, annotators begin by selecting a designated task from the

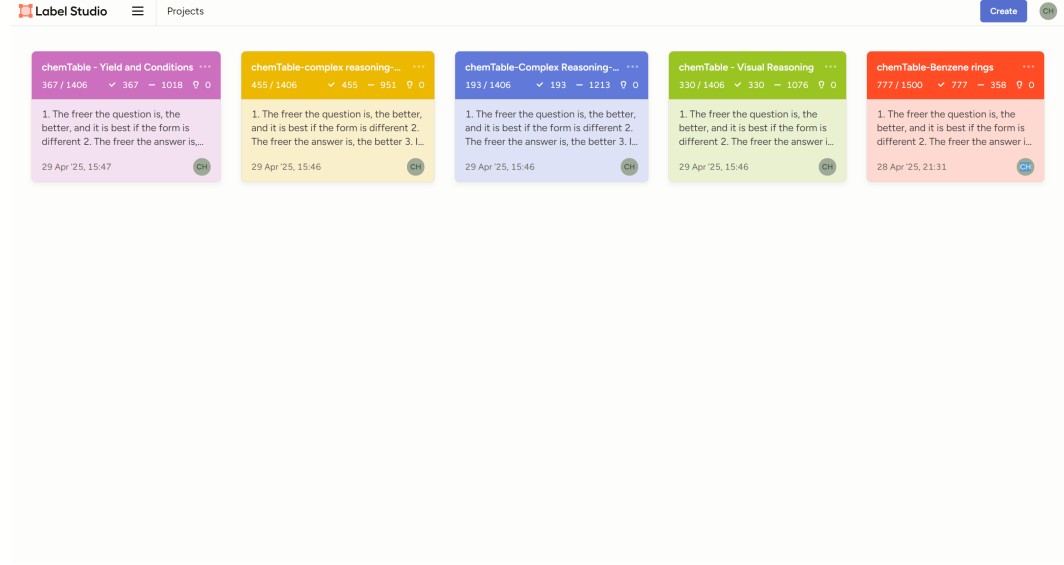

Figure 10: LabelStudio Dashboard View Showing Task Categories for Question-Answer Annotation.

project dashboard, where each task corresponds to a specific QA category (e.g., *Yield and Conditions*, *Visual Reasoning*, *Benzene Rings*).

Once a task is selected, the annotator is presented with the annotation interface (Figure 11), which displays the target table image at the top. Annotators must first select a suitable question type label. Based on the selected label, they are required to design a corresponding question-answer pair by analyzing the table content. To aid consistency, a reference panel on the right side of the interface provides example questions tailored to the selected type.

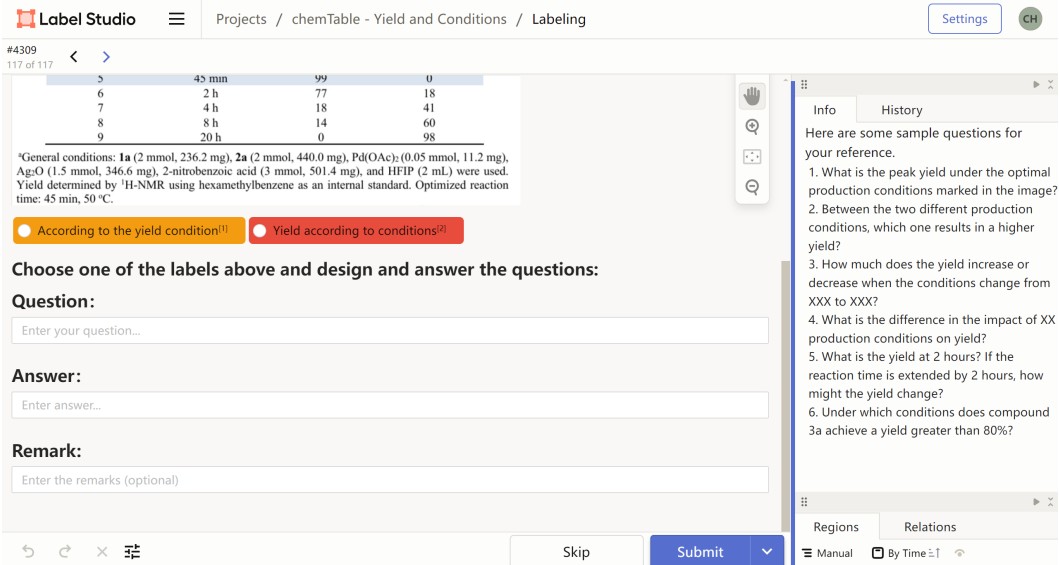

Figure 11: Annotation Interface for Generating QA Pairs from Table Content: Annotators Select a Question Type, Input the Question and Answer, and Refer to Provided Examples on the Right.

We encourage free-form question formulation to improve diversity. However, we enforce a strict constraint: all answers must be either directly found in the table or logically inferable from it without requiring specialized chemical knowledge. This ensures that questions remain grounded and accessible. In addition, annotators are encouraged to include **unanswerable questions** when the table content is ambiguous or insufficient. Such cases must be clearly noted in the *Remark* field to support subsequent filtering or diagnostic analysis.

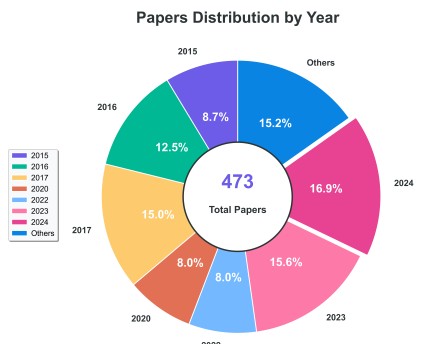

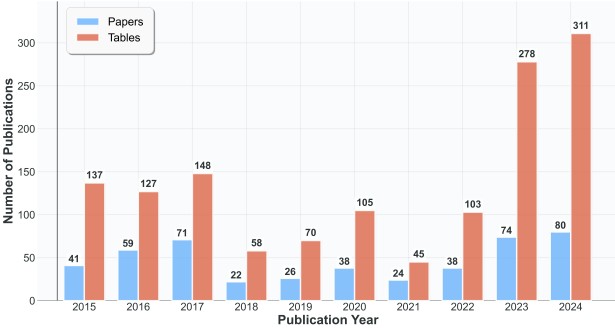

Figure 12: Year-wise distribution of papers in the dataset.

Figure 13: Temporal distribution of papers and extracted tables across the dataset.

## I  DATASET DISTRIBUTION AND CHEMICAL DIVERSITY

The dataset was curated from leading chemistry journals between 2015 and 2024, ensuring both disciplinary relevance and temporal breadth. The majority of tables originate from Angewandte Chemie International Edition, Organic Letters, The Journal of Organic Chemistry, ACS Catalysis, and JACS, reflecting their central role in reporting experimental results. The year-wise distribution shows a steady increase in table usage, with recent years contributing the largest share, consistent with the growing trend of structured data reporting in chemical research. The detailed distribution can be found in Figures 12 and 13.

Table 7: Scaffold and reaction-type diversity statistics of the dataset.

| Dimension | Metric | Value / Evidence |
|---|---|---|
| Scaffold diversity | Unique Bemis–Murcko scaffolds | 839 |
| | Scaffold-to-molecule ratio | 0.208 |
| | Mean pairwise Tanimoto similarity | 0.095 |
| | Top-3 scaffolds (share) | Benzene 19.7% · No-ring 15.1% · Cyclohexane 10.9% |
| Reaction-type coverage | Distinct reaction classes | 15 |
| | Top-3 classes (share) | C–C bond formation 16.4% · Oxidation 14.1% · C–Heteroatom bond 11.6% |

Beyond bibliometric statistics, the dataset exhibits substantial chemical diversity. We identified 839 unique Bemis–Murcko scaffolds, yielding a scaffold-to-molecule ratio of 0.208 and a low mean pairwise Tanimoto similarity (0.095). The most frequent scaffolds include benzene (19.7%), acyclic frameworks (15.1%), and cyclohexane (10.9%), together covering less than half of the molecules. Reaction coverage spans 15 distinct classes, dominated by C–C bond formation (16.4%), oxidation (14.1%), and C–heteroatom bond formation (11.6%).

Taken together, these distributions highlight both the representativeness of the literature sources and the structural and functional diversity of the chemical space, making the dataset a robust testbed for benchmarking multimodal models in chemistry.

## J  ANNOTATOR INFORMATION AND CONSISTENCY

The annotations for this study were performed by a team of chemistry experts with graduate-level education in the field. Their extensive training ensures a deep understanding of chemical terminology, experimental procedures, and domain-specific knowledge, which is crucial for the accurate interpretation and annotation of complex chemical tables.

To measure the consistency and reliability of the annotations, several quality control metrics were employed. The inter-annotator agreement (IoU) for cell boundaries reached 0.96, while the exact

match accuracy for SMILES extraction was 0.99. Additionally, the inter-annotator agreement for cell content text was 0.94, further confirming the high consistency of the annotations across different annotators.

These quality control measures underscore the reliability of the annotated data set, making it suitable for subsequent analyzes and model evaluations in chemical table recognition and understanding tasks.

## K EVALUATING DOMAIN-SPECIFIC AND GENERAL TABLE MODELS ON CHEMTABLE TASKS

This section evaluates the performance of domain-specific and general table models on chemistry-related tasks. Specifically, we compare ChemVLM (chemical domain-specific model) and Table-LLaVA 1.5 (general table model) in the context of chemical table understanding. The performance results across various question types are summarized in Figure 14.

The results indicate that general table models, such as Table-LLaVA, show limited transferability to chemistry tasks, primarily due to their lack of adaptation to the symbolic and multimodal nature of chemical tables. On the other hand, ChemVLM, which is specialized for the chemistry domain, performs better on certain tasks but still faces significant challenges in structured understanding and reasoning, especially with complex chemical data.

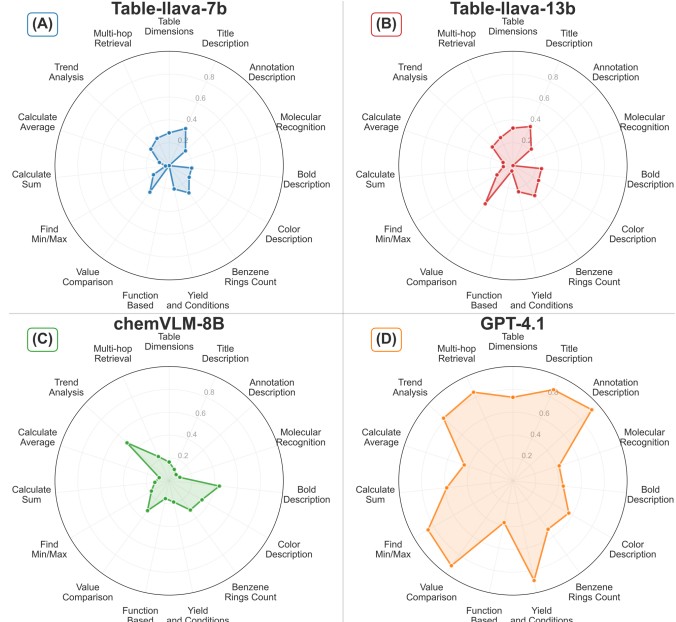

Figure 14: Performance comparison of domain-specific and general table models on ChemTable tasks.

These findings highlight a substantial gap in model performance when it comes to understanding and processing chemical tables in scientific literature. While ChemVLM and Table-LLaVA perform well in their respective domains, they still fall significantly short of models like GPT-4.1 in addressing the unique challenges of chemical table comprehension. ChemTable serves as a benchmark that reveals these gaps, offering a realistic and challenging testbed for advancing model capabilities in chemical table recognition and reasoning.

## L IMPLEMENTATION DETAILS

To ensure consistency and stability across different models during evaluation, we adopted a unified decoding configuration for all multimodal large language models. Specifically, the temperature was set to 0.0 and the nucleus sampling parameter (top-p) to 0.2, wherever supported. This configuration minimizes randomness and promotes deterministic outputs, which is essential for fair and reproducible evaluation across both table recognition and question answering tasks.

Our human performance results are based on the annotations of five chemistry domain experts, all with graduate-level training. To ensure fairness, annotators were provided only with scratch paper and a basic calculator without advanced functions and were instructed to answer the given questions directly. We used a balanced assignment scheme in which each question was independently answered by three randomly selected annotators, and reported human performance as the average accuracy across their responses. Human performance is only reported for tasks that require chemical expertise or complex reasoning; purely descriptive element-level tasks are not annotated by humans.

# M   QUALITATIVE CASE STUDIES

We present several representative failure cases that reveal where current MLLMs struggle when processing chemical tables. These cases span four primary error types identified in our benchmark—(1) fine-grained recognition failures, (2) missing visual-style grounding, (3) domain-specific chemistry errors, and (4) multi-hop reasoning failures—and are further supplemented by an integrated example in the final, where multiple failure modes co-occur within a single real-world table. Together, these qualitative analyzes provide a more complete view of the limitations observed in both recognition and reasoning tasks.

## M.1   FINE-GRAINED RECOGNITION ERRORS

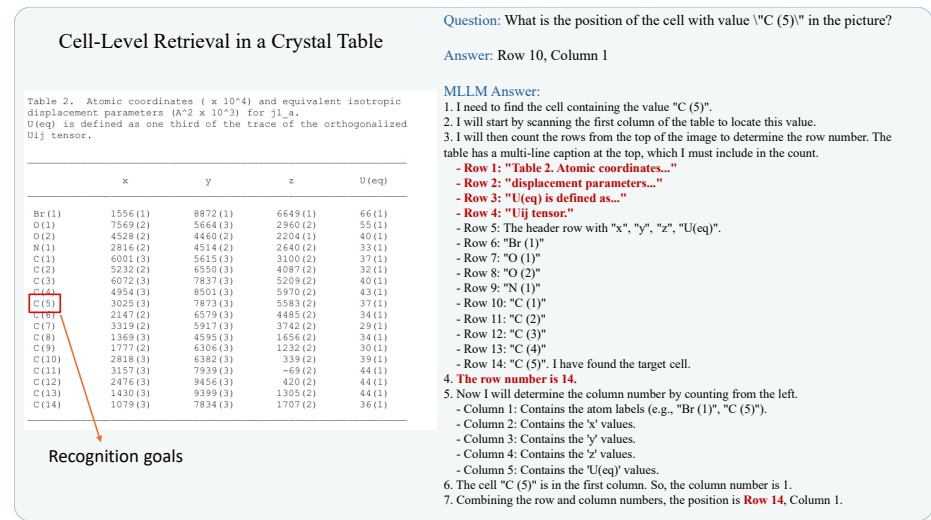

Figure 15: Case study of a fine-grained recognition failure on cell-level position retrieval: Gemini 2.5 Pro correctly OCRs and highlights the cell C(5) in a crystallographic coordinate table, but miscounts caption and header lines as data rows.

In Figure 15, we illustrate a typical fine-grained recognition error on the position-retrieval task. The table reports atomic coordinates from a crystallographic study, and the model is asked: *"What is the position of the cell with value 'C(5)' in the picture?"* While Gemini 2.5 Pro correctly OCRs the entry 'C(5)' and visually localizes it in the leftmost column, it still outputs an incorrect row index. The chain of thought shows that it sequentially numbers every line of the multi-line caption as table rows, then treats the header row as an additional row before starting to count the body entries. As a result, the true location "Row 10, Column 1" is misreported as "Row 14, Column 1," even though the target cell itself is clearly identified. This case demonstrates that fine-grained structural alignment—mapping correctly recognized content to precise row–column indices—remains a key bottleneck, even when overall layout parsing and OCR quality appear to be strong.

## M.2   VISUAL-STYLE GROUNDING FAILURES

In Figure 16, we probe visual-style grounding with a descriptive QA prompt posed to GPT-5: "What is the header of the column with yellow color?" The ground-truth answer is "No cells matching this characteristic exist," since the table uses only uniform grey shading to group yield columns and contains no yellow highlighting. GPT-5 nevertheless hallucinates a "colored" column, describing a shaded strip with 100% yields and confidently concluding that its header is 3a. This error shows that the model loosely maps any subtle background tint to the queried color instead of verifying the exact pixel-level style. Together with the large human–model gap on bold/color description in Table 4, this case indicates that even state-of-the-art MLLMs like GPT-5 still lack robust grounding of color and formatting cues in chemical tables: they can generate fluent style-aware explanations, but these explanations are often visually unsubstantiated.

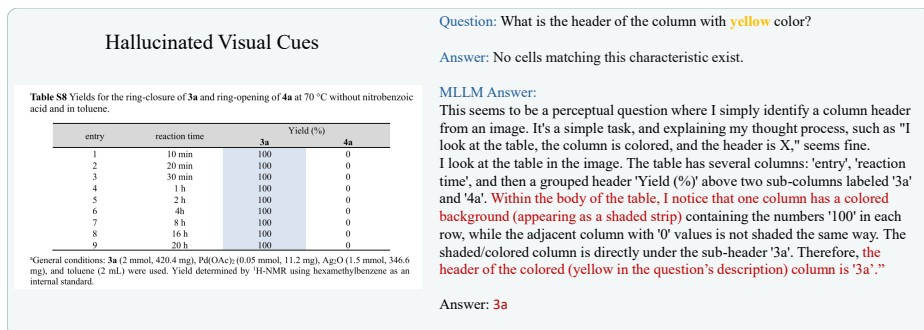

Figure 16: Case study of visual-style grounding failure. The model forces an alignment between a query for a "yellow" column and a blue-shaded column, illustrating a failure in negative verification of visual attributes.

## M.3 DOMAIN-SPECIFIC SYMBOL & REPRESENTATION CHALLENGES

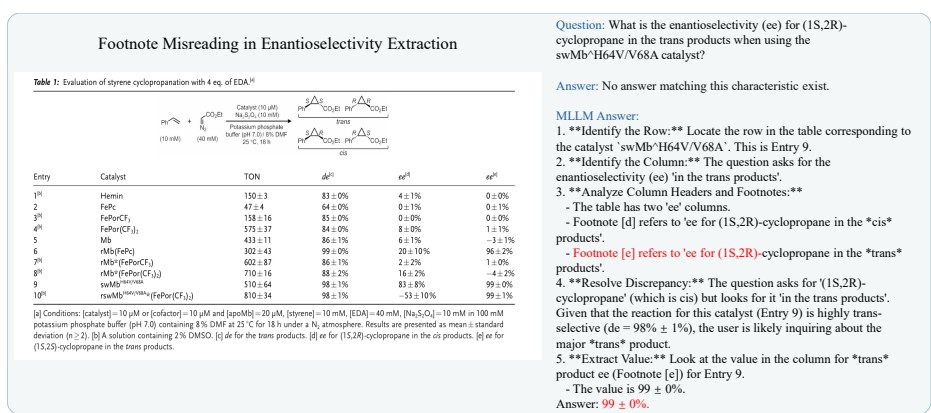

Figure 17: Case study of a domain-specific notation failure: Claude-4.5-Sonnet misinterprets stereo-chemical footnote labels in an enantioselectivity table.

In Figure 17, we highlight how chemistry-specific notation can derail table understanding even for strong MLLMs like Claude-4.5-Sonnet. The table reports enantioselectivities for several cyclopropane diastereomers, with superscript footnotes tying each column to a particular stereoisomer. The model is asked: "What is the enantioselectivity (ee) for (1S,2R)-cyclopropane in the trans products when using the swMdo H64V/V68A catalyst?" The correct answer is that no such value is reported, because the relevant footnote actually refers to a different stereochemical assignment than the queried (1S,2R) trans product. Claude nonetheless treats footnote [e] as if it matched the target species, propagates this mislink through its chain-of-thought, and confidently returns "99% ee." This case shows that current models still struggle to reliably bind domain-specific symbols and footnotes to their precise chemical meaning, and tend to hallucinate plausible numerical answers rather than recognizing when a requested quantity is formally undefined in the table, and refuse to answer this question.

## M.4 MULTI-HOP REASONING BREAKDOWNS

In Figure 18, we stress-test multi-hop reasoning in Claude-4.5-Sonnet with a query that first requires locating the row where S4 and S5 have yields of 20% and 35%, and then returning the corresponding Entry index. The model successfully reads all numerical values, correctly identifies that only Entry 4 satisfies both constraints, and even states in its chain-of-thought that "Entry 4 matches both criteria exactly." However, instead of outputting the requested index "4," it reports the base used in that row, LDA (2.0 M in THF/EtPh/heptanes), effectively answering a different column than the one specified in the question. This mismatch between correct intermediate reasoning and the final prediction suggests that the bottleneck is not local retrieval or numerical comparison, but the final hop that

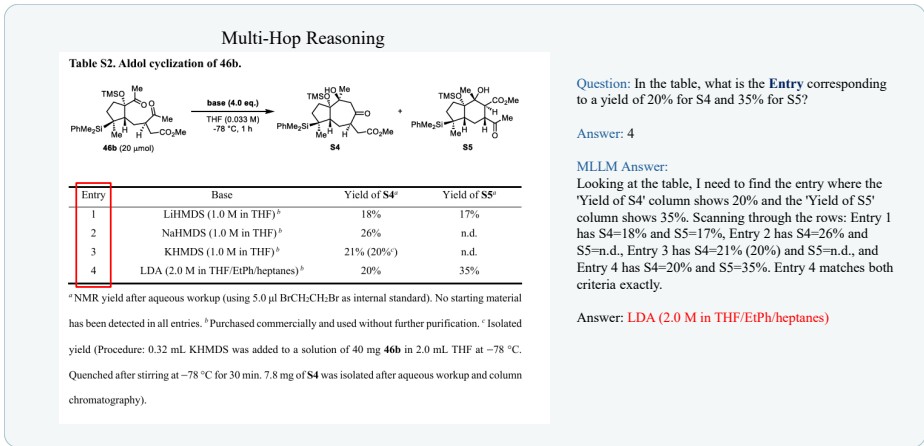

Figure 18: Case study of a multi-hop reasoning failure where Claude-4.5-Sonnet locates the correct row satisfying the yield constraints but outputs the base instead of the requested entry index.

maps a resolved row back to the correct schema field—an error pattern we observe repeatedly across multi-hop tasks in ChemTable.

## M.5 INTEGRATED CASE ACROSS DESCRIPTIVE AND REASONING TASKS

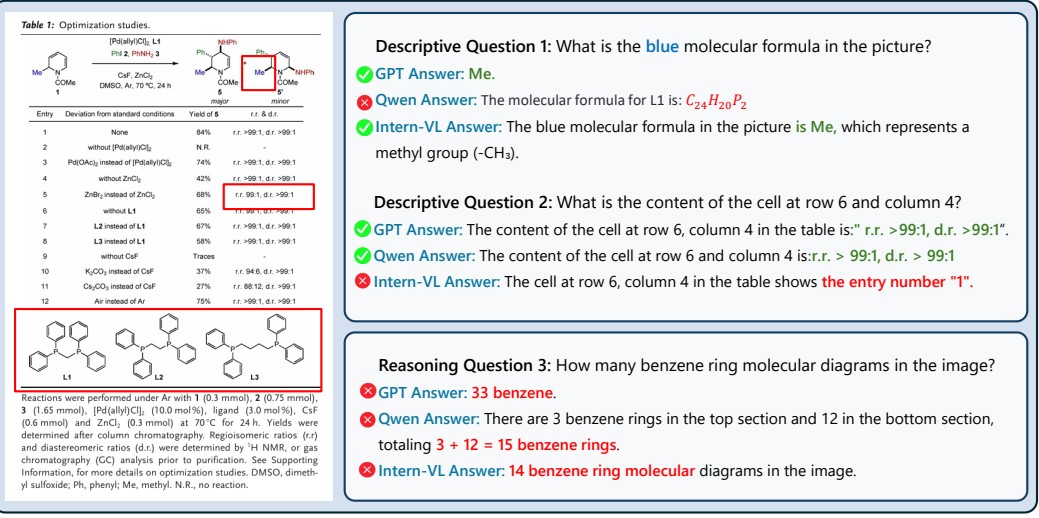

Figure 19: Case study of answering descriptive and reasoning questions with different MLLMs.

Figure 19 provides a compact example illustrating how multiple failure modes can emerge within a single chemical table. For a simple descriptive query about the blue-labeled "Me" group, some models correctly interpret the colored annotation, while others misidentify it as the molecular formula of a nearby ligand, reflecting unstable grounding of domain-specific visual symbols. For a cell-level retrieval question, certain models also mislocate the target cell despite accurate OCR.

When the task shifts to chemically grounded reasoning—such as counting benzene rings in the ligand structures—all evaluated models produce large errors. This underscores the limitations highlighted: even strong MLLMs struggle with visual chemistry reasoning, especially in structures containing repeated or fused aromatic motifs. Overall, this example shows how symbolic misinterpretation, positional errors, and domain reasoning failures can co-occur within the same table context.

## N    QA Density and Category Distribution in ChemTable

We provide a detailed analysis of QA density in ChemTable to ensure that evaluation is not dominated by a small subset of tables or QA types. The final filtered split used for all experiments contains 9,886 QA pairs over 1,382 tables (7,344 descriptive + 2,542 reasoning QAs). The per-table QA density is moderate: the mean is 7.2 QAs per table (median 7, minimum 1, maximum 18, inter-quartile range 5–9). Only a small fraction of tables are very heavily annotated (2% of tables have 16 questions), while over 80% of tables lie in the 5–15 QA range. The resulting Gini coefficient over the "QAs per table" distribution is 0.18, indicating a well-balanced and relatively uniform distribution. As shown in Table 8, the largest descriptive categories (Value Retrieval and Position Retrieval) each contribute only about 15–16% of all QAs. Reasoning-oriented categories such as Yield & Conditions, Multi-hop Retrieval, and Numerical Statistics are also well represented.

Table 8: Question–answer density and category statistics of the dataset.

| Dimension | Metric | Value / Evidence |
|---|---|---|
| Overall QA density | Tables with QA | 1,382 tables |
| | QA instances (total) | 9,886 QAs (7,344 descriptive / 2,542 reasoning) |
| | Mean / median QA per table | 7.2 / 7 |
| | Min / max QA per table | 1 / 18 |
| | 25–75th percentile (QA per table) | 5–9 |
| | Gini coefficient (QA density) | 0.18 (lower is more uniform) |
| QA count per table | 1–5 QAs per table | 410 tables (29.7% of 1,382) |
| | 6–10 QAs per table | 710 tables (51.4% of 1,382) |
| | 11–15 QAs per table | 230 tables (16.6% of 1,382) |
| | ≥16 QAs per table | 32 tables (2.3% of 1,382) |

## O    Dataset Visualization by Image Type

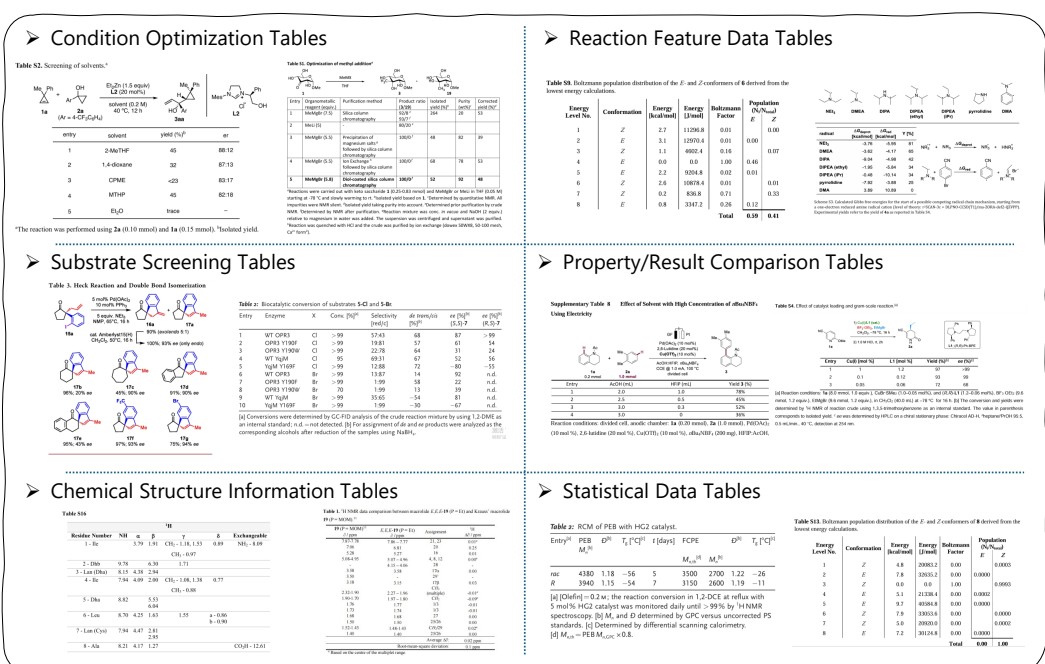

Figure 20: Representative Examples of Six Chemical Table Types in the ChemTable Dataset.

## P    TAXONOMY OF QUESTION TYPES WITH REPRESENTATIVE CASES

Table 9: Representative Question Across Different Task Types for Chemical Table Understanding.

| **Question Type** | Question Case 1 | Question Case 2 | Question Case 3 |
|---|---|---|---|
| Table Dimensions | What is the size of the table in the picture? | - | - |
| Title Description | What is the title of this table? | - | - |
| Annotation Description | What are the annotations of this table? | - | - |
| Visual Description | What is the reaction time in the row highlighted in light blue? | In the catalyst column, what is the content in bold? | For the row highlighted in light blue, which has a higher yield, 3a or 4a? |
| Benzene Rings Count | How many molecular diagrams of benzene rings are there in the table? | How many benzene rings are in the diagram? | What is the proportion of substances containing benzene rings among all the substances in the table? |
| Yield and Conditions | What is the reaction time when the yield is at its highest? | Under the condition of 50°C, at what reaction time is the yield highest? | At a temperature of 70°C and a reaction time of 30 minutes, which has a higher yield, 3a or 4a? |
| Function Based | What is the yield (%) of 3f at the reaction time where the yield first drops below half of its maximum value? | At which entry does the yield of 3f become less than the yield observed at 30 min, and what is the yield at that entry? | Which structure has the highest number of solvent atoms, and what is that number? |
| Numerical Statistics | What is the mean (average) value of I0/I? | At 223 K, which is higher: the calculated $v_a - v_b$ or the observed $v_a - v_b$? | What is the sum of the yields of product 2 across all solvents? |
| Trend Analysis | How does the yield of 3p change with increasing reaction time? | What is the trend in the third column? | As k increases, what is the trend in the change of obs.$v_a - v_b$? |
| Multi-hop Retrieval | Which entries in the table have a yield of 99? | What is the maximum value in the U(eq) column of the table? | In the figure, what is the reaction time corresponding to a 10% yield of 6c? |

# Q PROMPT TEMPLATES

## Q.1 TABLE RECOGNITION PROMPT TEMPLATES

Prompt 1 is designed to evaluate a model's ability to extract and reconstruct the structural layout of a table from an input image. The instruction explicitly requests the HTML representation of the table using only five basic tags: <table>, <thead>, <tbody>, <tr>, and <td>. To ensure a semantically accurate output, the prompt emphasizes the separation of the table header and body using <thead> and <tbody> tags, respectively. This task tests the model's understanding of both visual layout and hierarchical table semantics without reliance on style or advanced formatting. It serves as a foundational prompt for assessing table structure recognition capabilities.

---

**Prompt 1**

**Instruction:** Identify the structure of the table and return it to me in HTML format.
{image}

**Note:** 1. Use the <thead> and <tbody> tags to distinguish the table header from the table body.

2. Use only five tags: <table>, <thead>, <tr>, <td>, and <tbody>.

**Answer:**

---

Figure 21: HTML Table Structure Identification from Images.

Prompt 2 focuses on cell-level retrieval by requiring the model to locate the exact position of a cell within a table image based on a given content string. The model must identify the table structure, search for the specified cell content, and return the corresponding row and column indices in a structured JSON format.

---

**Prompt 2**

**Instruction:** Please identify the table in the picture and retrieve the corresponding cell position according to the cell content given and output.
{image}, {cell_content}

**Note:** 1. The cell position is represented by the row and column numbers of the cell. The row and column numbers start from 1.
2. If the cell is not found, please return an empty string.
3. All operations are performed on the entire table, including the head and body. When counting the number of rows and columns, the header and body of the table are also counted. If there are headers and columns, The row count starts from the header row. The columns count starts from the header columns.
4. Your answer must be returned in the following json format.

{"chain_of_thought": "Your thought process for complete this task.",
"row_index": 0,"col_index": 0}

**Answer:**

---

Figure 22: Locate Cell Position by Content in Table Images.

Prompt 3 evaluates the model's ability to locate and extract the textual content of a specific cell based on given row and column indices within a table image. The coordinates are one-indexed and cover both the header and body of the table. The model must parse the structure visually, identify the correct cell, and return its string content in a JSON object.

---

**Prompt 3**

**Instruction:** Please identify the table in the picture and retrieve the corresponding cell according to the row and column coordinates given and output. {image}, {cell_index}

**Note:** 1. The coordinates of the cell are represented by the row and column numbers of the cell. The row and column numbers start from 1.
2. The cell value is a string, and the output format is a string.
3. If the cell is empty, please return an empty string.
4. All operations are performed on the entire table, including the head and body. When counting the number of rows and columns, the header and body of the table are also counted. If there are headers and columns, The row count starts from the header row. The columns count starts from the header columns.
5. Your answer must be returned in the following json format.

{"chain_of_thought": "Your thought process for complete this task.","content": "The cell content."}

**Answer:**

---

Figure 23: Retrieve Cell Content by Position in Table Images.

Prompt 4 targets the task of molecular recognition by instructing the model to identify molecular structures in a given image and convert them into SMILES (Simplified Molecular Input Line Entry System) format. This task evaluates a model's capacity for visual parsing of chemical diagrams, structural interpretation, and chemical knowledge alignment. The expected output is a valid SMILES string encapsulated within <smiles> tags, ensuring format consistency.

---

**Prompt 4**

**Instruction:** Identify the molecules in this image and return them to me in smiles format.

**Format:** The answer is wrapped in <smiles></smiles>.

**Answer:**

---

Figure 24: Molecular Recognition and SMILES Conversion from Images.

## Q.2 TABLE RECOGNITION PROMPT TEMPLATES

Prompt 5 evaluates a model's ability to extract the title text from a scientific document image. The expected output is a JSON object containing both the extracted title and a brief explanation outlining the reasoning process.

---

**Prompt 5**

**Instruction:** Extract the title from this chemical document image.

**Format:** Return the results in the following JSON format:
```json
{
    "chain_of_thought": "your chain of thought about how you get the
final result.",
    "title": "Title text"
}
```

**Answer:**

---

Figure 25: Extract Document Title from Image.

Prompt 6 asks the model to answer a question by reasoning over an HTML-rendered table. Given the structured table content and a specific question, the model must provide the answer to the question.

---

**Prompt 6**

**Instruction:** Please answer the question based on the html content of the table.

**Table:** {Table_html}

**Format:**
```json
{
    "chain_of_thought": "your chain of thought about how you get the
final result.",
    "answer": "answer"
}
```

**Question:** {Question}

**Answer:**

---

Figure 26: Answer Table-Based Question from HTML Structure.

Prompt 7 requires the model to identify the total number of rows and columns in a table image. The output is formatted as a JSON object with a reasoning chain and a count of rows and columns. The task evaluates the model's capability in structural table parsing and its consistency in counting visual elements in scientific layouts.

---

**Prompt 7**

**Instruction:** Please identify the table in the picture, and retrieve the dimensions of the table and output.
{image}

**Note:** 1. The dimensions of the table are represented by the number of rows and columns.

2. The dimensions of the table are two integers. The row and column numbers start from 1.

3. All operations are performed on the entire table, including the head and body. When counting the number of rows and columns, the header and body of the table are also counted. If there are headers and columns, The row count starts from the header row. The columns count starts from the header columns.

4. Your answer must be returned in the following json format.

{"chain_of_thought": "Your thought process for complete this task.", "rows": 0,"columns": 0}

**Answer:**

---

Figure 27: Retrieve Table Dimensions from Image.

Prompt 8 challenges the model to answer a natural language question based on a table image. This tests the model's vision reasoning ability by requiring joint understanding of the image content and question intent.

---

**Prompt 8**

**Instruction:** Please answer the question based on the image of the table.

**Format:**
```json
{
    "chain_of_thought": "your chain of thought about how you get the final result.",
    "answer": "answer"
}
```

**Question:** {Question}

**Answer:**

---

Figure 28: Visual Table Question Answering.

Prompt 9 evaluates the model's ability to answer questions using both a table image and its text. By combining unstructured (visual) and structured (HTML) inputs, this prompt tests how effectively the model integrates both modalities to improve accuracy and handle noise or ambiguity in either format.

---

**Prompt 9**

**Instruction:** Please answer the question based on the html content of the table and the image of the table.

**Table:** {Table_html}

**Format:**
```json
{
  "chain_of_thought": "your chain of thought about how you get the final result.",
  "answer": "answer"
}
```

**Question:** {Question}

**Answer:**

---

Figure 29: Multimodal Table QA with HTML and Image Inputs.

Prompt 10 asks the model to produce five question–answer pairs in statistical categories (e.g., max, sum, mean, compare) based on a table image and its HTML content.

---

**Prompt 10**

**Instruction:** Based on the table in the picture and the content of the table, generate 5 statistical questions (compare, find the maximum value, calculate the sum and mean of values). Please give the question-answer pair. Return it to me in json format.
category:
1. compare
2. max
3. sum
4. mean

**Table:** {Table_html}

**Format:**
```json
{
  "chain_of_thought": "your chain of thought about how you get the final result.",
  "QA": [{"question": "question 1","answer": "answer 1","category": "..."},{"question": "question 2","answer": "answer 2","category": "..."}, ... ]
}
```

**Answer:**

---

Figure 30: Generate Statistical Questions and Answers from Table.

Prompt 11 requires the model to assess whether a given answer correctly responds to a question. It must return a binary decision ("correct" or "incorrect") and explain its reasoning.

---

**Prompt 11**

**Instruction:** Please evaluate the answer based on the question and the answer. If the answer is correct, please return "correct". If the answer is incorrect, please return "incorrect".
If the answer is unable to answer the question, please make sure the model's answer is refused to answer the question.

**Question:** {Question}

**Answer:** {Answer}

**Model's Answer:** {Model_Answer}

**Format:**
```json
{
    "chain_of_thought": "your chain of thought about how you get the final result.",
    "is_correct": "correct or incorrect"
}
```

**Answer:**

---

Figure 31: Answer Evaluation and Judgement Prompt.

Prompt 12 tasks the model with summarizing the content and purpose of a scientific table, based on its HTML structure and visual appearance. The expected summary should concisely capture the data's meaning, key variables, and scientific implications.

---

**Prompt 12**

**Instruction:** Based on the table in the picture and the content of the table, please generate a concise summary that explains the main purpose and content of the table. The summary should include what the table is about, what research or data it presents, and its key findings or implications.

**Table:** {Table_html}

**Format:**
```json
{
    "chain_of_thought": "your chain of thought about how you get the final result.",
    "summary": "summary"
}
```

**Answer:**

---

Figure 32: Table Summarization for Scientific Contexts.

## R    THE USE OF LARGE LANGUAGE MODELS

We utilized large language models (LLMs) to assist and enhance the preparation of this manuscript. Specifically, LLMs were employed to improve clarity, grammar, and readability, while all conceptual, benchmark methodological, and experimental contributions are original and developed by the authors.

## S    COPYRIGHT AND LICENSING

The dataset presented in this work was constructed by collecting and processing table images extracted from published scientific articles. We ensured that all source articles fall under licenses that permit such use, including **CC0**, **CC-BY 4.0**, **CC-BY-SA 4.0**, and **CC-BY-NC 4.0**. For each extracted table image, we have provided explicit attribution to the original publication, including its DOI.

The journals from which images were sourced include:

- *Science* (`https://www.science.org/journal/science`)
- *Chem* (`https://www.sciencedirect.com`)
- *Journal of the American Chemical Society* (`https://pubs.acs.org/journal/jacsat`)
- *ACS Catalysis* (`https://pubs.acs.org/journal/accacs`)
- *Angewandte Chemie International Edition* (`https://onlinelibrary.wiley.com/journal/15213773`)
- *Organic Letters* (`https://pubs.acs.org/journal/orlef7`)

After annotation and compilation, the resulting dataset is released under the **Creative Commons Attribution-ShareAlike 4.0 International (CC BY-SA 4.0)** license, which permits reuse, redistribution, and adaptation, provided appropriate credit is given and any derivative works are licensed under the same terms. Our code is licensed under the **Apache License 2.0**. All table images are subject to the copyright terms of their original publications and publishers.

