# OpenReview forum: "Benchmarking Multimodal LLMs on Recognition and Understanding over Chemical Tables"
_ICLR.cc/2026/Conference — Submitted to ICLR 2026_

### Official Review · Reviewer_zKp5 · 2025-10-30

**Soundness:** 2
**Presentation:** 2
**Contribution:** 2
**Rating:** 4
**Confidence:** 3

**Summary:**

The paper introduces ChemTable, a large-scale benchmark for evaluating multimodal large language models (MLLMs) on chemical table recognition and understanding. The dataset includes 1,382 real-world chemical tables extracted from top-tier chemistry journals, annotated with structure, content, and over 9,000 QA pairs. The authors evaluate both open-source and proprietary MLLMs on tasks such as table structure extraction, molecular recognition, and reasoning over chemical data. The key claim is that current MLLMs fall short of human-level performance, especially in domain-specific reasoning and molecular structure understanding.

**Strengths:**

This paper presents ChemTable, a carefully constructed and richly annotated benchmark focused on chemical table recognition and understanding, which fills a notable gap in the evaluation of multimodal large language models (MLLMs) on domain-specific scientific content. The dataset is derived from real-world chemistry literature and includes over 1,300 tables and nearly 10,000 question-answer pairs, supporting both structural extraction and reasoning tasks. The authors provide a comprehensive evaluation pipeline and assess a wide range of open-source and proprietary models, offering a useful reference for the community. The work is clearly written, well-organized.

**Weaknesses:**

- The experimental conclusions lack depth and novelty. The finding that proprietary models outperform open-source ones is widely acknowledged and not specific to this domain. The paper does not provide detailed error analysis or insights into why models fail, making the conclusions too generic to guide future model development.

- The paper does not sufficiently justify the necessity of a chemistry-specific benchmark in light of existing general-purpose benchmarks like MMMU or HLE, which also include chemistry-related content. There is no comparative analysis showing that ChemTable introduces uniquely challenging or uncovered tasks, weakening the motivation for a new dataset.

- The evaluation metrics, while standard, are not diagnostic. High-level accuracy scores do not reveal whether models truly understand chemical content or are relying on superficial cues. There is no attempt to evaluate intermediate reasoning steps or semantic correctness, especially in molecular recognition and multi-hop reasoning tasks.

- The paper lacks exploration of training strategies or model behavior. All models are evaluated in a zero-shot or frozen setting, with no investigation into whether domain-specific pretraining or fine-tuning improves performance. This limits the benchmark’s utility as a tool for driving model development rather than just evaluation.

- The generalizability of the findings is not discussed. While the benchmark is chemistry-specific, there is no attempt to assess whether insights or improvements from ChemTable transfer to other scientific domains or table types, limiting its broader impact.

**Questions:**

Please see weaknesses.

---

> ### Author Response · Authors · 2025-11-20
> **Official Response to Reviewer zKp5 [1/3]**
>
> Thank you for this thoughtful and constructive review! We genuinely appreciate your expertise in evaluating ChemTable and the perceptive questions that have helped us strengthen our work. Let us address each of your valuable points in detail.
>
> > W1: Could the experimental analysis be strengthened with more detailed error analysis and insights into model failure patterns, beyond the general finding that proprietary models outperform open-source ones?
>
> Thank you for this valuable comment. We agree that simply showing proprietary models outperforming open-source ones would not be novel. Our goal, however, is to go beyond that. In ChemTable, we analyze why models fail in the chemistry-table domain. We isolate factors such as molecular-structure density, symbolic notation, and fine-grained cell alignment, and show that all models—regardless of openness—break down sharply when these domain-specific elements appear. We further examine multi-hop complexity, forward vs. inverse condition–yield reasoning, and hybrid (text+image) input sensitivity, which reveal concrete reasoning asymmetries that do not appear in general-domain benchmarks.
>
> In the revised version, we have expanded Section 5 and Appendix M with new qualitative and quantitative case studies. These highlight recurring failure modes—including incorrect mapping between molecules and their row/column indices, hallucinated reaction structures, misinterpretation of symbolic shorthand, and last-step extraction errors despite correct evidence retrieval. By dissecting these patterns, our work provides actionable insights for designing models that better integrate visual, symbolic, and structural cues in scientific documents. Ultimately, we believe these findings can help guide more targeted model training, better multimodal alignment strategies, and the construction of future models that reason more faithfully over scientific data.

---

> ### Author Response · Authors · 2025-11-20
> **Official Response to Reviewer zKp5 [2/3]**
>
> > W2: How does ChemTable differ from existing general-purpose benchmarks like MMMU or HLE that include chemistry-related content? Comparative analysis showing unique challenges would strengthen the motivation for this new dataset.
>
> We thank the reviewer for this question and agree that broad multimodal benchmarks such as MMMU and HLE are crucial for general evaluation. However, these datasets are explicitly designed as exam-style academic benchmarks, with only a small subset involving chemistry and very limited use of real experimental tables. In contrast, ChemTable targets a different setting: all examples are tables extracted from recent peer-reviewed chemistry literature, with cell- and molecule-level annotations (logical/physical coordinates, titles, footnotes, reaction schemes, mapped SMILES, etc.). Our goal is to assess whether a model can function as a chemistry-aware “research assistant” that understands and operates on real tables in papers, rather than as a generic exam solver.
>
> | Aspect                        | ChemTable                                                                                          | MMMU                                                                                           | HLE                                                                                          |
> | ----------------------------- | -------------------------------------------------------------------------------------------------- | ---------------------------------------------------------------------------------------------- | -------------------------------------------------------------------------------------------- |
> | Primary goal                 | Benchmark chemistry-specific **table recognition & understanding** from real literature           | General **multi-discipline multimodal** exam-style benchmark                                  | Frontier-level **academic exam** for multimodal LLMs                                          |
> | Data source                  | **extracted from peer-reviewed chemistry papers** (2015–2024) with DOIs         | multimodal questions from **college exams, quizzes, textbooks** across many subjects     | expert-written questions forming a **“last exam”** across dozens of disciplines      |
> | Domain focus                 | **Chemistry** esp. reaction/optimization/substrate-scope & property tables                  | Broad multi-discipline , small chemistry subset | Broad multi-discipline , small chemistry subset |
> | Basic unit                   | **Full experimental table** with reaction schemes, molecular diagrams, footnotes, etc.            | **Standalone exam question** + associated image(s) (charts, tables, chemical structures, …)   | **Standalone exam question** (MCQ/short answer), sometimes with a figure                     |
> | Table / document structure   | Fine-grained **cell logical & physical coordinates**, titles, notes, reactions, substances        | Some images are tables, but **no cell-level table structure or reaction-table annotation**     | No real experimental tables; **no table structure annotations**                              |
> | Chemistry-specific signals   | Explicit **reaction schemes, molecules mapped to SMILES**, domain abbreviations, footnotes, etc.  | A few questions involve chemistry diagrams, but **no SMILES-level or table-centric supervision** | Chemistry appears only as **high-level academic questions**, not real experimental tables    |
> | Task types                   | **Table recognition** (structure reconstruction, value/position retrieval, molecular recognition) + **table QA** (descriptive, numerical, logical, domain-specific) | Closed-ended multimodal **QA** (mainly exam-type questions) requiring college-level reasoning | Closed-ended **multimodal QA** (MCQ + short answer) targeting frontier academic difficulty   |
> | Target use case / scenario   | **Chemistry research assistant**: understanding and operating on tables in real papers            | **General exam taker**: broad academic multimodal reasoning                                   | **Ultimate academic exam**: benchmark LLMs against human experts on exam problems            |
>
> Beyond this difference in data source and use case, ChemTable introduces task types that are not covered in MMMU/HLE: table recognition tasks such as value retrieval, position retrieval, and molecular recognition (diagram → SMILES), and table understanding tasks that require integrating symbolic notations, molecular graphics, visual styles, and quantitative outcomes (e.g., benzene-ring counting, yield-vs-condition reasoning, function-based QA). Our results show that even state-of-the-art MLLMs remain far from human performance on these chemistry-specific abilities, revealing a capability gap that general exam-style benchmarks do not expose.

---

> ### Author Response · Authors · 2025-11-20
> **Official Response to Reviewer zKp5 [3/3]**
>
> > W3: Could more diagnostic evaluation metrics reveal whether models truly understand chemical content rather than relying on superficial cues? Evaluating intermediate reasoning steps or semantic correctness would be valuable.
>
> Thank you for this valuable suggestion. We agree that deeper diagnostic analysis of model reasoning is an important direction.
>
> Our current evaluation primarily uses accuracy-based metrics across diverse task types, and we believe this already reveals meaningful insights about model limitations. Even with this straightforward metric, ChemTable successfully exposes systematic failures of state-of-the-art MLLMs on chemistry tables—particularly on tasks requiring fine-grained molecular recognition, symbolic notation understanding, and multi-step reasoning. The fact that these gaps emerge across all model families suggests fundamental limitations in how current MLLMs process structured scientific content.
>
> To complement these quantitative results and address your suggestion, we have added qualitative case studies in Appendix M that examine the intermediate reasoning processes of flagship models. These case studies visualize the step-by-step predictions, including retrieved cells, predicted SMILES, and chain-of-thought reasoning, to illustrate concrete failure patterns—such as localization errors, visual-style confusion, domain-specific symbol misinterpretation, and multi-hop breakdown. While our primary contribution remains the benchmark and the quantitative evaluation, these initial analyses help clarify *where* and *why* models fail and point toward promising directions for future research on improving multimodal scientific reasoning.
>
> > W4: Would exploring training strategies beyond zero-shot evaluation, such as domain-specific pretraining or fine-tuning, enhance the benchmark's utility for driving model development?
>
> Thank you for highlighting the lack of domain-specific pretraining or fine-tuning experiments. Our primary goal in this work is to introduce ChemTable as a diagnostic, model-agnostic benchmark that reveals systematic failure modes of current MLLMs on realistic chemical tables, rather than to advocate a particular training recipe. By evaluating models in a zero-shot setting, we aim to expose core bottlenecks in multimodal alignment and domain-grounded reasoning—such as interpreting molecular structures, handling symbolic notations, and following implicit layout conventions—which more faithfully reflect the models’ inherent generalization capabilities. While fine-tuning on domain-specific samples could boost performance on this specific dataset, it potentially compromises the model's generalization ability across diverse scientific tasks.
>
> Regarding the benchmark’s role in driving model development, we see careful characterization of current limitations as a prerequisite to designing better training strategies. In particular, our analysis (including the comparison between general models such as Table-LLaVA and chemistry-oriented models such as ChemVLM in Appendix K) indicates that simply injecting domain data does not resolve deeper structural and semantic challenges in chemical table understanding. ChemTable is intended to make these gaps explicit so that future work on domain-specific pretraining, fine-tuning regimes, and architectural changes can be targeted at clearly identified weaknesses rather than optimized only for aggregate benchmark scores. We will clarify this design rationale and its implications for model development in the revised version.
>
> > W5: How might insights from ChemTable transfer to other scientific domains or table types, and could this broader transferability be assessed to demonstrate wider impact?
>
> Thank you for raising the question of generalizability beyond chemistry-specific tables. Our choice of chemical tables is deliberate: they form an extreme yet representative instance of scientific tables that combine dense symbols, complex layouts, and multimodal graphics, so the failure modes we expose (e.g., weak fine-grained retrieval, sensitivity to symbolic notations, difficulty with visual styling, numerical reasoning gaps, and handling of unanswerable queries) stem from generic limitations of current MLLMs rather than chemistry alone. Moreover, the core components of ChemTable—the annotation schema with logical/physical coordinates and style tags, the separation of recognition vs. understanding, the hybrid text+image evaluation setting, and the automatic QA protocol—are designed to be domain-agnostic and can be directly reused to build analogous benchmarks for other scientific domains (e.g., biology, materials science, or financial/statistical tables).
>
> ---
>
> Your insightful feedback has been invaluable in refining our work and clarifying key aspects of ChemTable. Should you have any further thoughts or wish to explore any aspect in greater detail, we would be delighted to continue this productive discussion.

---

### Official Review · Reviewer_v8z9 · 2025-11-01

**Soundness:** 2
**Presentation:** 3
**Contribution:** 3
**Rating:** 4
**Confidence:** 4

**Summary:**

This paper presents ChemTable. It is a large-scale benchmark designed to evaluate the recognition and understanding abilities of MLLMs when interacting with real-world chemical tables.
ChemTable includes more than 1300 tables collected from top chemistry journals.
Each table is enriched with pixel-level annotations, logical layout structures, and domain-specific labels.
The benchmark supports two major tasks: table recognition (structure parsing and content extraction) and table understanding (descriptive and reasoning QA).
The authors conduct extensive experiments on both proprietary and open-source MLLMs. They benchmark these models across a diverse set of tasks and compare their ability with human performance.

**Strengths:**

- This paper makes a valuable contribution by introducing a benchmark on chemical tables, an area previously lacking multimodal datasets.
Unlike prior table benchmarks (e.g., FinQA, SciTab, MMTab), ChemTable focuses on chemistry-specific content that includes molecular structures, chemical symbols, and experimental conditions.
- The dataset contains over 1300 real-world chemistry tables and 9000 question–answer pairs, covering a wide range of tasks such as table recognition, structure parsing, and reasoning-based question answering.
The annotation process is comprehensive, including layout, text, molecular graphics, and metadata.
- The authors conduct an extensive comparison across multiple leading open-source (e.g., Qwen-VL, Llama, InternVL) and proprietary models (e.g., GPT-4.1, Gemini-2.5-Flash).
This benchmarking provides a clear and balanced view of current model capabilities and limitations in chemical table understanding.

**Weaknesses:**

- The core evaluation framework of ChemTable mainly adopts existing standard metrics such as TEDS, Edit Distance, and Accuracy. While these metrics are reliable, they are not tailored to capture the unique characteristics of chemical tables (e.g., molecular structures, symbolic notation, multimodal relationships). The evaluation approach appears to be a direct transfer from general table recognition tasks, lacking methodological innovation specific to the chemistry domain.
- As a benchmark, ChemTable focuses almost exclusively on general-purpose multimodal large language models (MLLMs), while lacking systematic evaluation of domain-specific models for chemistry or broader scientific applications. This omission limits its value as a professional reference benchmark for “chemical table” understanding and recognition.
- Many tasks within the benchmark (such as Table Recognition, Title Description, Annotation Description, Yield and Conditions, Value Comparison, Find Min/Max, and Multi-hop Retrieval) already achieve high performance across mainstream MLLMs, with most models scoring above 80. This raises a key concern: if general models already perform well on these tasks, do these benchmarks still provide sufficient discriminative power or research value?
- Notably, Gemini-2.5-Flash, a small-sized model, achieves the best or near-best performance in most tasks, which raises an important question: if a “smaller” model performs this well, would flagship models such as Gemini-2.5-Pro, GPT-5, or Claude-4 easily surpass the current results? If so, does the benchmark still provide meaningful evaluation or differentiation between stronger models?

**Questions:**

Please refer to the weakness part above.

---

> ### Author Response · Authors · 2025-11-20
> **Official Response to Reviewer v8z9 [1/2]**
>
> We deeply appreciate your meticulous review and the valuable feedback you have provided. Your detailed observations and suggestions have significantly contributed to strengthening our research. We are committed to addressing each of your concerns comprehensively below.
>
> > W1: Could the evaluation framework benefit from metrics more tailored to chemical tables' unique characteristics (molecular structures, symbolic notation, multimodal relationships) rather than primarily using standard metrics like TEDS, Edit Distance, and Accuracy?
>
> Thank you for the insightful comment. Our choice of TEDS and accuracy is intentional: one of ChemTable's goals is to serve as a broadly accessible benchmark for the community, and these metrics are the established standards in table recognition and QA, ensuring comparability, reproducibility, and ease of interpretation for non-chemistry researchers. Importantly, we did tailor the evaluation to the chemical domain. As stated in Sec. 4.1 (p.6), for cells containing molecular structures, we replace the normalized edit distance with the Tanimoto coefficient—chemoinformatics' gold standard for assessing molecular similarity—thereby addressing the unique challenges of molecular graph recognition.
>
> Beyond metrics, our evaluation framework incorporates domain-specific task designs: symbolic interpretation is captured through dedicated descriptive QA tasks (e.g., Bold/Color Description), and multimodal relational reasoning is evaluated through complex reasoning tasks such as Function-Based QA, which require jointly understanding text, symbols, reaction conditions, and molecular diagrams. We highlight these domain-aware components more clearly in the revised version.
>
> > W2: Would systematic evaluation of domain-specific chemistry models and broader scientific applications strengthen ChemTable's value as a professional reference benchmark?
>
> We appreciate your suggestion to emphasize domain-specific models. We would like to clarify that our benchmark does include evaluations of specialized models specifically designed for chemistry and scientific tables. As detailed in Appendix K (Figure 14), we evaluated ChemVLM, a multimodal model finetuned specifically for chemistry tasks, comparing it against general table models (e.g., Table-LLaVA). Additionally, in Section 4.2 (Figure 3), we evaluated DECIMER, a specialized model for molecular structure recognition, to provide a baseline for fine-grained recognition tasks.
>
> Our analysis reveals that while ChemVLM outperforms general table models on chemistry-centric questions (e.g., tasks involving molecular entities and reaction conditions), it still faces significant challenges in complex structured reasoning compared to strong general baselines. This finding validates ChemTable’s value as a rigorous reference point, as it exposes the limitations that persist even in domain-adapted models. In the revised manuscript, we will move the discussion of these domain-specific baselines from the Appendix to the main text to ensure the systematic evaluation of scientific models is more prominent.

---

> ### Author Response · Authors · 2025-11-20
> **Official Response to Reviewer v8z9 [2/2]**
>
> > W3: Given that many tasks achieve high performance (>80%) across mainstream MLLMs, do these benchmarks still provide sufficient discriminative power or research value for advancing the field?
>
> We appreciate your concern about the relatively high scores achieved by mainstream MLLMs on several ChemTable tasks. Our design philosophy is to provide a complete and unbiased evaluation of table understanding in realistic chemical settings. The "easier" tasks (e.g., table recognition, basic title/annotation description, yield-and-conditions lookup) are essential in real scientific workflows and serve as sanity checks and anchor points. They allow us to verify that models can robustly handle foundational skills under non-trivial domain shift and diagnose where performance starts to break down along the pipeline. Moreover, even on these sub-tasks, models are not saturated: we still observe noticeable gaps and different error patterns across model families, which are informative for understanding model strengths and weaknesses.
>
> Most importantly, the core challenge and discriminative power of ChemTable lie in the chemistry-specific and compositional tasks—such as molecular recognition, benzene-ring counting, function-based QA, visual-style interpretation, and arithmetic aggregation—where state-of-the-art models remain far below human experts. These high-headroom tasks, together with the "easy-but-essential" ones, make ChemTable a long-term, high-ceiling benchmark for tracking genuine progress in multimodal scientific reasoning.
>
> > W4: Given the strong performance of Gemini-2.5-Flash, how might flagship models like Gemini-2.5-Pro, GPT-5, or Claude-4 perform on ChemTable? Understanding the performance ceiling would help clarify the benchmark's discriminative capacity.
>
> We appreciate this insightful question regarding the ceiling of our benchmark. To address this, we extended our evaluation to include the strongest available flagship models: GPT-5, Gemini-2.5-Pro, and Claude-4.5-Sonnet (results added to Table 4). Our findings indicate that while these flagship models improve upon smaller models like Gemini-2.5-Flash, they do not saturate the benchmark, and ChemTable remains a rigorous testbed that successfully differentiates between top-tier capabilities.
>
> **Table: Performance comparison of latest models on ChemTable**
>
> | Task                   | GPT-5 | Gemini-2.5-Pro | Claude-4.5-Sonnet |
> |------------------------|:-----:|:--------------:|:-----------------:|
> | Table Dimensions       | 74.89 | 74.35          | 76.11             |
> | Title Description      | 83.74 | 87.67          | 87.30             |
> | Annotation Description | 93.11 | 89.91          | 87.41             |
> | Molecular Recognition  | 52.04 | 69.31          | 58.14             |
> | Bold Description       | 40.53 | 45.81          | 48.93             |
> | Color Description      | 50.78 | 54.56          | 48.19             |
> | Benzene Rings Count    | 57.22 | 63.67          | 46.00             |
> | Yield and Conditions   | 90.53 | 93.69          | 90.81             |
> | Function Based         | 37.94 | 73.97          | 45.66             |
> | Value Comparison       | 86.44 | 92.00          | 91.85             |
> | Find Min/Max           | 86.18 | 89.62          | 94.79             |
> | Calculate Sum          | 60.65 | 56.43          | 46.32             |
> | Calculate Average      | 47.84 | 50.00          | 50.85             |
> | Trend Analysis         | 84.46 | 87.32          | 86.21             |
> | Multi-hop Retrieval    | 83.68 | 84.87          | 85.65             |
>
> Specifically, significant performance gaps remain between flagship models and human experts, particularly in domain-intensive tasks. This confirms that ChemTable captures the nuanced limitations of even the most powerful MLLMs and remains a valuable tool for measuring progress in scientific multimodal understanding.
>
> ---
>
> Thank you once again for your rigorous assessment and constructive critique. Your feedback has been instrumental in refining our methodology and highlighting important aspects of our work. We look forward to any additional questions or suggestions you might have, and we remain eager to engage in further discussion.

---

### Official Review · Reviewer_K6DN · 2025-11-01

**Soundness:** 3
**Presentation:** 3
**Contribution:** 2
**Rating:** 6
**Confidence:** 3

**Summary:**

The paper proposes a benchmark for evaluating multimodal LLMs on chemical table recognition and understanding. The dataset spans ~1,300 annotated chemical tables from literature with 9000 QA instances to measure table understanding capabilities. The work evaluates multiple MLLMs on the benchmark, revealing performance gaps in molecular structure recognition and domain reasoning compared to human performance.

**Strengths:**

Novel benchmark focused on a challenging domain of chemical table recognition and reasoning

Rigorous annotational protocol incorporating both manual and synthetic data generation

Comprehensive evaluation across multiple open-source and proprietary MLLMs

**Weaknesses:**

Limited dataset scale. Although the benchmark incorporates multiple table types, the overall number of samples remains modest, with 41.4% of the tables related to “Condition Optimization”. These limitations can constrain the generalizability of the findings. It would be good if the authors can provide some discussion of sampling bias and coverage across chemistry subfields.

Questions distribution. The paper does not report the distribution of QA instances across tables. Given that multiple filtering steps are applied, how did the authors ensure that evaluation metrics are not biased by skewed QA density? It would be helpful to include per-table or per-category QA statistics after filtering.

Qualitative error analysis. Although the paper provides metrics for different subdomains of the benchmark (e.g. by question type, molecular complexity), it would be helpful to include a few concrete failure case studies with visualizations to clarify where the models break.

**Questions:**

I'm not so sure about the contribution—the authors propose a dataset for OCR in chemistry; overall, it seems useful, but not for the broader community. Looks like a paper for Chemoinformaics journals.

---

> ### Author Response · Authors · 2025-11-20
> **Official Response to Reviewer K6DN[1/2]**
>
> We are grateful for your supportive evaluation and recognition of our work's contribution to the community. Your constructive feedback has been invaluable in strengthening our research. Below, we address your specific suggestions to further improve our work.
>
> > W1: Could the current dataset scale and the concentration on "Condition Optimization" tables (41.4%) potentially limit the generalizability of findings? Additional discussion on sampling bias and coverage across chemistry subfields would strengthen the work.
>
> Thank you for the insightful comment. We emphasize that our objective is to curate high-quality, expert-level chemical tables rather than to maximize sample count, as the fidelity of structural and semantic annotations is essential for benchmarking multimodal reasoning. The distribution of table types is intentional rather than a sampling artifact: as clarified in Sec. 3.1.1 and Appendix D.3, condition-optimization and substrate-screening tables are the dominant reporting formats in experimental chemistry and naturally constitute the majority of tables in contemporary publications. Regarding coverage across chemistry subfields, we ensured broad disciplinary representation by sourcing data from a diverse set of top-tier journals—including ACS Catalysis, JACS, Chem, Angew. Chem. Int. Ed., and Science—which collectively span organometallic catalysis, synthetic methodology, organic synthesis, photochemistry, electrochemistry, and reaction mechanism studies. As summarized in Appendix M, our dataset includes six major functional table categories (optimization, scope, structural information, reaction-feature data, property comparison, and statistical tables), each represented by >10% of the dataset. This demonstrates that ChemTable captures a wide range of chemical research practices and subfields rather than being confined to a narrow experimental domain.
>
> > W2: How can we ensure evaluation metrics are not biased by skewed QA density, given the multiple filtering steps applied? Including per-table or per-category QA statistics after filtering would be valuable.
>
> We appreciate the opportunity to elaborate on the QA density. While Table 2 outlines the dataset scale, we now provide the specific distribution across tables on the final filtered split used for all experiments. This split contains **9,886 QA pairs over 1,382 tables** (7,344 descriptive + 2,542 reasoning QAs). The per-table density is moderate: the mean is **7.2** QAs per table (median **7**, min **1**, max **18**, inter-quartile range **5–9**). Only a small fraction of tables are very heavily annotated: as summarized below, **≈2%** of tables have ≥16 questions, while over **80%** of tables lie in the 5–15 QA range. The resulting Gini coefficient over the “QAs per table” distribution is **0.18**, indicating a well-balanced and relatively uniform distribution.
>
> | Statistic                     | Value   |
> | ----------------------------- | ------- |
> | # tables with QA              | 1,382   |
> | # QA instances (total)        | 9,886   |
> | Mean / median QA per table    | 7.2 / 7 |
> | Min / max QA per table        | 1 / 18  |
> | 25–75th percentile            | 5 – 9   |
> | Gini coefficient (QA density) | 0.18    |
>
>
>
> | QA count per table | # tables (% of 1,382) |
> | ------------------ | --------------------- |
> | 1–5 QAs            | 410 (29.7%)           |
> | 6–10 QAs           | 710 (51.4%)           |
> | 11–15 QAs          | 230 (16.6%)           |
> | ≥16 QAs            | 32  (2.3%)            |
>
> To further check that evaluation is not dominated by a few tables or QA types, we also report the per-category QA counts for the same filtered split, aligning with the taxonomy in **Table 4** and **Table 8** of our paper. The largest descriptive categories (Value and Position Retrieval) each contribute only about **15–16%** of all QAs. Crucially, the reasoning tasks (e.g., Yield & Conditions, Multi-hop, Statistics) are well-represented, and no single category dominates the dataset. Thus, the reported metrics reflect a balanced evaluation across the diverse capabilities described in Section 3.3.
>
> | QA Category (Task Name) | # QAs | Share of all QAs |
> | :--- | :--- | :--- |
> | **Value Retrieval** (Descriptive) | 1,537 | 15.5% |
> | **Position Retrieval** (Descriptive) | 1,538 | 15.6% |
> | **Title Description** | 1,296 | 13.1% |
> | **Table Dimensions** | 1,256 | 12.7% |
> | **Annotation Description** | 957 | 9.7% |
> | **Benzene Rings Count** | 729 | 7.4% |
> | **Multi-hop Retrieval** | 428 | 4.3% |
> | **Numerical Statistics** (Sum, Avg, etc.) | 412 | 4.2% |
> | **Unanswerable** (Format/Style issues) | 371 | 3.8% |
> | **Yield and Conditions** | 344 | 3.5% |
> | **Visual Description** (Color/Bold) | 310 | 3.1% |
> | **Function Based / Domain Specific** | 292 | 3.0% |
> | **Unanswerable** (Ambiguity/No Answer) | 235 | 2.4% |
> | **Trend Analysis** | 181 | 1.7% |

---

> ### Author Response · Authors · 2025-11-20
> **Official Response to Reviewer K6DN[2/2]**
>
> > W3: While the paper provides metrics across different subdomains, concrete failure case studies with visualizations would help clarify where models encounter difficulties.
>
> We thank the reviewer for highlighting the importance of qualitative error analysis. We completely agree that, beyond quantitative metrics across question types and molecular complexity, it is very helpful to show concrete visual failure cases to clarify where models break. In the revised manuscript, we have added a dedicated qualitative error-analysis section with visualized case studies (Appendix M). For each case, we present the original ChemTable image, the ground-truth answer, and the model’s full prediction, so that readers can see step by step how the model’s reasoning diverges from the correct one rather than only looking at aggregate scores.
>
> In these new case studies, we organize and illustrate several recurring failure patterns: (1) fine-grained localization errors, where a model correctly reads a cell (e.g., a specific substituent or entry ID) but misaligns it with the row/column indices and returns a neighbor entry; (2) missing visual-style grounding, where the model hallucinates or confuses bold/colored regions and therefore selects the wrong column; (3) domain-specific chemistry mistakes, such as misinterpreting stereochemical symbols or footnotes and inferring values that are not actually reported; and (4) multi-hop breakdowns, where the model finds the correct row satisfying certain yield or condition constraints but outputs the wrong field in the final step. We explicitly link each visual example to the corresponding ChemTable sub-tasks (e.g., molecular recognition, benzene-ring counting, function-based QA, and multi-hop retrieval), so that the qualitative analysis directly explains and complements the quantitative patterns you observed. We hope these additions address your suggestion and make it much clearer where current models fail on ChemTable.
>
> > Q1: Given that you propose a domain-specific dataset, how can you demonstrate that your work provides value to the broader deep learning community, beyond just being a resource for the chemoinformatics field?
>
> We appreciate your concern and welcome the opportunity to clarify the scope of our contribution. While the task involves OCR, we want to clarify that ChemTable is designed to benchmark Multimodal Reasoning, not just recognition. It jointly evaluates table recognition and table understanding, requiring fine-grained multimodal alignment, symbolic parsing, numerical reasoning, and domain-grounded inference—capabilities central to modern MLLMs. We use chemistry as a stress test, not an end domain: chemical tables combine text, symbols, formatting cues, and graphical molecular structures, forming an unusually challenging but representative setting for scientific multimodal reasoning. Our experiments show that state-of-the-art general-purpose MLLMs consistently fail on these tasks (e.g., visual-text alignment, robust arithmetic, domain-informed reasoning), revealing fundamental weaknesses not exposed by existing general-domain benchmarks. We believe ChemTable provides the ICLR community with a needed resource for advancing robust multimodal understanding in real-world scientific scenarios.
>
> ---
>
> We sincerely appreciate your thoughtful comments, which have helped us clarify the scope and impact of our work. If you have further questions or suggestions, we would be very happy to address them.

---

### Official Review · Reviewer_84uM · 2025-11-01

**Soundness:** 2
**Presentation:** 3
**Contribution:** 2
**Rating:** 4
**Confidence:** 3

**Summary:**

The paper proposes ChemTable, a benchmark consists of chemical tables with various contents. The core tasks include table recognition and table understanding. Evaluations are carried out for several closed-source or open-source multi-modal models, showing that they consistently fall short in complex tasks such as handling molecular structures and symbolic conventions.

**Strengths:**

The authors construct a real-world chemical table benchmark with expert annotations. The evaluated tasks are diverse, and the paper presents adequate number of experimental observations that seems reasonable.

**Weaknesses:**

Major points:

* The size of benchmark seems to be inadequately large.
* The evaluated models are not state-of-the-art. What are the performance of more powerful models, such as Claude-4.0, GPT-5, Gemini-2.5-Pro? Also, the paper should include MLLMs specifically finetuned for chemistry tasks, such as ChemVLM.
* The presentation form of tables is always image. Did you try out tabular data form and study the performance of relational foundation models? Or using text descriptions for symbolic elements / graph representations for molecular structures when applicable?
* This is a purely benchmarking paper, and the authors fail to provide theoretical justifications or insights in depth. In particular, for those open-sourced models where one can observe the reasoning patterns inside the models, can you provide any analysis?

Minor points:

* Typo: "Claude-3-7-Sonnet" should be "Claude-3.7-Sonnet" in Table 3.

**Questions:**

See above

---

> ### Author Response · Authors · 2025-11-20
> **Official Response to Reviewer 84uM[1/2]**
>
> We sincerely thank you for your thorough evaluation and constructive feedback on our work. Your insights have been instrumental in helping us improve the quality and clarity of our paper. We address each of your concerns in detail below.
>
> > W1: Could the current benchmark size be sufficient for comprehensive evaluation?
>
> We appreciate this important consideration regarding the benchmark scale. Our design philosophy prioritizes quality over quantity to ensure high-fidelity, domain-representative evaluation. Chemical tables present unique challenges—combining molecular structures, symbolic conventions, and complex visual layouts—requiring expert-driven, labor-intensive annotation. This careful curation results in a rigorous benchmark that better reflects real-world scientific reasoning demands. Nonetheless, we agree that scale is valuable, and we are expanding ChemTable, including enlarging the benchmark, adding cross-table QA, and continuously maintaining and extending the dataset to support broader scientific multimodal evaluation.
>
>
> > W2: Would expanding the evaluation to include more recent state-of-the-art models and domain-specific chemistry models strengthen the benchmark's comprehensiveness?
>
> Thank you for these valuable suggestions. We have substantially expanded our experiments to include stronger multimodal models from all three major families. In the revised version, we now report results for GPT-5, Gemini-2.5-Pro, and Claude-4.5-Sonnet on ChemTable (updated Table 4). These models improve over earlier baselines but still fall clearly short of human experts across the main ChemTable sub-tasks. This confirms that ChemTable remains challenging even for the latest MLLMs and that there is still a noticeable gap to expert-level performance. We also plan to release a continuously updated leaderboard to track the performance of emerging MLLMs on ChemTable.
>
> **Table: Performance comparison of latest models on ChemTable**
>
> | Task                   | GPT-5 | Gemini-2.5-Pro | Claude-4.5-Sonnet |
> |------------------------|:-----:|:--------------:|:-----------------:|
> | Table Dimensions       | 74.89 | 74.35          | 76.11             |
> | Title Description      | 83.74 | 87.67          | 87.30             |
> | Annotation Description | 93.11 | 89.91          | 87.41             |
> | Molecular Recognition  | 52.04 | 69.31          | 58.14             |
> | Bold Description       | 40.53 | 45.81          | 48.93             |
> | Color Description      | 50.78 | 54.56          | 48.19             |
> | Benzene Rings Count    | 57.22 | 63.67          | 46.00             |
> | Yield and Conditions   | 90.53 | 93.69          | 90.81             |
> | Function Based         | 37.94 | 73.97          | 45.66             |
> | Value Comparison       | 86.44 | 92.00          | 91.85             |
> | Find Min/Max           | 86.18 | 89.62          | 94.79             |
> | Calculate Sum          | 60.65 | 56.43          | 46.32             |
> | Calculate Average      | 47.84 | 50.00          | 50.85             |
> | Trend Analysis         | 84.46 | 87.32          | 86.21             |
> | Multi-hop Retrieval    | 83.68 | 84.87          | 85.65             |
>
>
> Regarding chemistry-specialized models such as ChemVLM, we fully agree they are important baselines. ChemVLM was already evaluated in our original submission as a chemistry-specific MLLM baseline (Appendix K, Fig. 14). Concretely, we compare ChemVLM with the general table model Table-LLaVA 1.5 on ChemTable. ChemVLM clearly outperforms Table-LLaVA on chemistry-centric questions (e.g., questions involving molecular entities and reaction conditions), but both models still perform substantially worse than GPT-4.1 and the strongest closed-source MLLMs, especially on structure-heavy tables with dense molecular graphs and multi-step reasoning. In the revised version, we highlight this in the main text and explicitly refer to the appendix.
>
>
> > W3: Could alternative input formats beyond images, such as structured tabular data and different representations for molecular structures, be explored?
>
> Thank you for this thoughtful suggestion. Our study has already extensively explored tabular data formats and evaluated relational foundation models across multiple modalities. As detailed in Section 5.3 and Figure 5, ChemTable supports text, image, and hybrid representations. For text-based settings, symbolic chemical elements are provided in canonical SMILES format, while in image-based settings molecular structures are presented directly as molecular graphs. Our experiments systematically compare these input forms and analyze their impact on recognition and reasoning performance, demonstrating that our benchmark rigorously evaluates models beyond image-only table formats.

---

> ### Author Response · Authors · 2025-11-20
> **Official Response to Reviewer 84uM[2/2]**
>
> > W4: Would deeper theoretical insights and analysis of model reasoning patterns, particularly for open-source models where internal mechanisms can be examined, strengthen this work?
>
> We thank the reviewer for this insightful comment and agree that a benchmark is most useful when it also explains *how* and *why* models succeed or fail. Beyond reporting overall scores, our work systematically analyzes the reasoning behavior of both proprietary and open-source MLLMs. We study how performance changes with reasoning complexity (multi-hop depth), and observe a consistent degradation as the number of required reasoning steps increases, highlighting compositional limitations. We further compare “forward’’ condition–yield queries with their “inverse’’ counterparts and find clear asymmetries: models are much more reliable when predicting yields from conditions than when inferring conditions from desired outcomes, suggesting that they mainly learn one-way correlations rather than invertible scientific relations.
>
> In the revised manuscript, we additionally include new qualitative case studies (Appendix M). These analyses reveal recurring failure modes—including misalignment between localized cells and row/column indices, hallucinated visual structure, incorrect binding of scientific symbols and footnotes, and last-step errors where models locate the right evidence but extract the wrong field—which directly explain the gap between strong structural understanding and weaker cell-level reasoning. Together, these quantitative analyses and qualitative case studies go beyond “pure benchmarking” and provide concrete insights into the reasoning patterns and failure mechanisms of models, as requested by the reviewer.
>
>
>
> > W5: There appears to be a minor typographical error in Table 3 that needs correction.
>
> Thank you for the careful reading and attention to detail. We have corrected "Claude-3-7-Sonnet" to "Claude-3.7-Sonnet" in Table 3 accordingly. We appreciate this helpful feedback.
>
> ---
>
> We greatly value your feedback, which has been crucial in enhancing our work. Your detailed comments and suggestions have helped us strengthen both the technical contributions and presentation of our paper. If you have any further questions or concerns, we would be delighted to discuss them with you.

---

### Author Response · Authors · 2025-11-20
**Overall Revision of the Submitted Manuscript**

We would like to thank all the reviewers for their efforts in reviewing our manuscript. Your valuable suggestions inspire us to improve our study and gracefully revise our manuscript. Specifically, we have resubmitted a newly revised manuscript following your suggestions. The main revision parts can be summarized as below.

We have updated the main comparison table to include three recent flagship MLLMs—GPT-5, Gemini-2.5-Pro, and Claude-4.5-Sonnet—and slightly refined our conclusions to reflect their improved performance while still highlighting the non-trivial gap to human experts on chemistry-intensive and compositional tasks. In addition, we have added a new Appendix M that presents qualitative case studies (with table images, ground-truth answers, and model predictions) to illustrate typical failure modes in localization, visual-style grounding, symbol/footnote binding, and multi-step reasoning. Finally, we have improved the overall exposition and organization of the paper, clarified several descriptions, and corrected minor typographical issues.

---

### Author Response · Authors · 2025-12-01
**Summary of Author Response**

We sincerely appreciate the opportunity to revise our work and are genuinely grateful to the reviewers for their thoughtful and constructive feedback. Given the changes in the discussion process, we would like to briefly summarize how our revision and rebuttals attempt to address the main concerns raised.

For clarity, we organize our responses by theme rather than by individual reviewer.

---

### 1. Dataset scale, diversity, and QA distribution

**Concern:** Is dataset large and diverse enough, and is the focus on “condition optimization” tables biased?

**What we clarified/added in the paper:**

* We emphasize that the goal is a **high-fidelity, expert-annotated** benchmark, not maximal size, and describe the expert-driven annotation protocol and agreement in more detail.
* We explain that optimization/scope tables are **indeed the dominant formats in modern experimental chemistry**, and also show that six functional table categories (optimization, scope, structural info, reaction-feature, property comparison, statistical tables) are each represented by >10%, covering multiple chemistry subfields.
* We add **QA-density statistics**: 9,886 QAs over 1,382 tables, with a mean of 7.2 QAs per table (median 7), and a balanced distribution across QA categories, so evaluation is not dominated by a few tables or an over-represented question type.

---

### 2. Model coverage and remaining headroom

**Concern:** Missing the strongest SOTA models; limited coverage of domain-specific models; risk that the benchmark is too easy or quickly saturated.

**What we clarified/added:**

* We **extend experiments** to include flagship models **GPT-5, Gemini-2.5-Pro, and Claude-4.5-Sonnet**. They improve over smaller baselines but still show clear gaps to human experts on chemistry-intensive and compositional tasks (e.g., molecular recognition, benzene-ring counting, function-based QA, and arithmetic aggregation).
* We highlight existing **chemistry-specific baselines** in the paper:

  * **ChemVLM** (chemistry-specialized MLLM) vs. a general table model (Table-LLaVA).
  * **DECIMER** for molecular-structure recognition.
    ChemVLM improves on chemistry-centric questions but still lags strong general MLLMs on complex reasoning.
* We report **human expert performance**, showing that ChemTable is far from saturated even for the strongest models.

---

### 3. Evaluation design, modalities, and relevance beyond “OCR.”

**Concern:** Metrics seem standard and not chemistry-specific; benchmark might be “OCR for chemistry”; unclear relation to general multimodal benchmarks.

**What we clarified/added:**

* We keep standard metrics (TEDS, edit distance, accuracy) for comparability, but make them **chem-aware** where needed (e.g., using **Tanimoto similarity over SMILES** for molecular cells) and explicitly analyze different types of **unanswerable questions** and model behavior on them.
* We stress that ChemTable supports **three input settings**: text-only (HTML), image-only, and **hybrid text+image**, and show that hybrid input improves key tasks like Yield & Conditions and function-based reasoning. Thus we benchmark both recognition and **multimodal understanding**, not just image OCR.
* We clarify the positioning relative to MMMU/HLE and prior table benchmarks: those contain exam-style or general-domain data, whereas ChemTable uses **real experimental chemistry tables with pixel-level, logical, and molecular annotations** and domain-grounded QA, targeting the “research assistant on real papers” use case.

---

### 4. Reasoning patterns and failure analysis

**Concern:** Need deeper insight into *why* models fail, beyond “proprietary > open-source”.

**What we clarified/added:**

* We add a **qualitative error-analysis section** with visual case studies, showing recurring failure modes:
  (i) misalignment between localized cells and indices,
  (ii) confusion about bold/color styles,
  (iii) misinterpretation of chemistry symbols/footnotes,
  (iv) last-step errors in multi-hop reasoning.
* We analyze how performance changes with **multi-hop depth**, molecular-structure density, and “forward vs inverse” condition–yield queries, revealing systematic compositional and correlation-vs-causality limitations that are relevant beyond chemistry.

---

We acknowledge that no work is perfect from the outset, and we are genuinely grateful for the reviewers' constructive and insightful feedback. We believe these revisions make meaningful progress in addressing the main concerns raised (on dataset design, model coverage, evaluation, and analysis), and substantially strengthen the clarity, rigor, and impact of the work.

---

### Meta-Review · Area_Chair_4kmC · 2025-12-20

**Summary:**

- Dataset Scope & Bias: Multiple reviewers (84uM, K6DN) questioned if the benchmark's size and its heavy focus on "Condition Optimization" tables were sufficient and representative, potentially limiting generalizability and indicating sampling bias.

- Model Coverage & Benchmark Ceiling: Reviewers (84uM, v8z9) noted the initial lack of evaluation on the strongest flagship models (GPT-5, Gemini-2.5-Pro, Claude-4.5) and domain-specific models (ChemVLM). There was concern that the benchmark might be too easy or quickly saturated, especially given the high performance of a smaller model like Gemini-2.5-Flash on many tasks.

- Depth of Analysis & Insight: Reviewers (84uM, zKp5, K6DN) felt the paper lacked deep, domain-specific error analysis and theoretical insights. The finding that proprietary models outperform open-source ones was seen as generic, and there was a call for more diagnostic evaluation and qualitative case studies to understand why models fail.

- Motivation & Broader Relevance: Reviewer zKp5 questioned the necessity of a chemistry-specific benchmark given existing general-purpose ones (MMMU, HLE). Reviewer K6DN also wondered about its value beyond the cheminformatics community, suggesting it might be more suited for a domain-specific journal.

- Evaluation Methodology: Reviewers (v8z9, zKp5) felt the evaluation metrics (TEDS, accuracy) were standard and not sufficiently tailored to the unique challenges of chemical tables (e.g., molecular structures). They suggested a need for more diagnostic or semantic metrics.

**Reviewer Concerns:**

- Evaluation Methodology (Partially Outstanding): While the qualitative analysis is a strong addition, the core quantitative metrics remain the standard ones. A reviewer deeply concerned about the lack of innovative, chemistry-tailored evaluation metrics might not be fully satisfied, though the practical need for comparability is a reasonable counter-argument.

- Generalizability Discussion (Partially Addressed): Reviewer zKp5's question about transferring insights to other domains was acknowledged and met with a theoretical argument about chemistry as an "extreme yet representative" case. While the authors state the framework is reusable, an explicit discussion of how the identified failure modes (e.g., symbol binding, fine-grained alignment) are generalizable to, say, biological or materials science tables would strengthen this point further.

**Reviewer Scores:**

- Reviewer 84uM: Originally 4 (marginally below threshold). The main concerns (model coverage, analysis depth) were directly addressed with new experiments and a dedicated error analysis section. Likely to increase score to 6 (above threshold).

- Reviewer K6DN: Originally 6 (marginally above threshold). Concerns about dataset balance and the need for qualitative analysis were met with new statistics and the new Appendix M. The question about broader relevance was answered satisfactorily. Likely to maintain or slightly increase confidence, solidifying a 6.

- Reviewer v8z9: Originally 4 (marginally below threshold). They have primary concerns about benchmark ceiling (addressed by new flagship model results) and domain-specific model evaluation. They might hold at a 4.

- Reviewer zKp5: Originally 4 (marginally below threshold). This was the most critical review, questioning novelty and depth. The rebuttal directly engaged with each point, adding significant analytical depth (error studies) and clarifying the unique niche vs. other benchmarks. This could persuade the reviewer, leading to a possible increase to a 5 (borderline/weak accept). However, if the reviewer remains unconvinced about the fundamental novelty or diagnostic value, they might hold at a 4.

The authors provide a lot of rebuttal, but the overall score is still lower than the ICLR bar. It is suggested to resubmit with a major revision.

---

### Decision · Program_Chairs · 2026-01-26

Reject